# Engineering unsymmetrically coordinated Cu-S$_1$N$_3$ single atom sites with enhanced oxygen reduction activity

Huishan Shang[1,13], Xiangyi Zhou[2,13], Juncai Dong [3,13], Ang Li[4], Xu Zhao[5], Qinghua Liu [5], Yue Lin [6], Jiajing Pei[7], Zhi Li[8], Zhuoli Jiang[1], Danni Zhou[1], Lirong Zheng[3], Yu Wang [9], Jing Zhou[9], Zhengkun Yang[10], Rui Cao[11], Ritimukta Sarangi[11], Tingting Sun[12], Xin Yang[2], Xusheng Zheng[5], Wensheng Yan [5], Zhongbin Zhuang [7], Jia Li [2✉], Wenxing Chen [1✉], Dingsheng Wang [8✉], Jiatao Zhang[1✉] & Yadong Li[8]

Atomic interface regulation is thought to be an efficient method to adjust the performance of single atom catalysts. Herein, a practical strategy was reported to rationally design single copper atoms coordinated with both sulfur and nitrogen atoms in metal-organic framework derived hierarchically porous carbon (S-Cu-ISA/SNC). The atomic interface configuration of the copper site in S-Cu-ISA/SNC is detected to be an unsymmetrically arranged Cu-S$_1$N$_3$ moiety. The catalyst exhibits excellent oxygen reduction reaction activity with a half-wave potential of 0.918 V vs. RHE. Additionally, through in situ X-ray absorption fine structure tests, we discover that the low-valent Cuprous-S$_1$N$_3$ moiety acts as an active center during the oxygen reduction process. Our discovery provides a universal scheme for the controllable synthesis and performance regulation of single metal atom catalysts toward energy applications.

[1] Beijing Key Laboratory of Construction Tailorable Advanced Functional Materials and Green Applications, School of Materials Science and Engineering, Beijing Institute of Technology, Beijing 100081, China. [2] Laboratory for Computational Materials Engineering, Division of Energy and Environment, Graduate School at Shenzhen, Tsinghua University, Shenzhen 518055, China. [3] Beijing Synchrotron Radiation Facility (BSRF), Institute of High Energy Physics, Chinese Academy of Sciences, Beijing 100049, China. [4] Beijing Key Laboratory of Microstructure and Property of Advanced Materials, Beijing University of Technology, Beijing 100029, China. [5] National Synchrotron Radiation Laboratory (NSRL), University of Science and Technology of China, Hefei 230029, China. [6] Hefei National Laboratory for Physical Sciences at the Microscale, University of Science and Technology of China, Hefei 230026, China. [7] State Key Lab of Organic-Inorganic Composites, Beijing University of Chemical Technology, Beijing 100029, China. [8] Department of Chemistry, Tsinghua University, Beijing 100084, China. [9] Shanghai Synchrotron Radiation Facilities (SSRF), Shanghai Institute of Applied Physics, Chinese Academy of Science, Shanghai 201204, China. [10] Collaborative Innovation Center of Chemistry for Energy Materials, Department of Chemistry, University of Science and Technology of China, Hefei 230026, China. [11] Stanford Synchrotron Radiation Lightsource, SLAC National Accelerator Laboratory, Menlo Park, CA 94025, USA. [12] Beijing Key Laboratory for Science and Application of Functional Molecular and Crystalline Materials, Department of Chemistry, University of Science and Technology Beijing, Beijing 100083, China. [13] These authors contributed equally: Huishan Shang, Xiangyi Zhou, Juncai Dong. ✉email: lijia@phys.tsinghua.edu.cn; wxchen@bit.edu.cn; wangdingsheng@mail.tsinghua.edu.cn; zhangjt@bit.edu.cn

The developing of advanced fuel cells and metal-air batteries equipped with oxygen electrodes provides new opportunities for the applications of future sustainable energy[1–3]. To realize energy conversion with highly efficiency, it's crucial to improve the oxygen reduction reaction (ORR) procedure, among these electrochemical devices[4,5]. Currently platinum-based materials have been widely used for ORR, but are unfortunately precluded by their rarity and high price[6]. Although the newly developed catalysts with earth-abundant elements exhibit some fancy properties, the overall performance including activity and durability is still far from satisfactory[7–10]. Hence, the rational design of ideal oxygen electrode materials with low-cost but high activity and good stability under applied conditions remains a formidable challenge.

Due to the high atomic utilization, single atom catalysts have gained great attention in heterogeneous catalysis, and significantly, they provide new horizons for the discovery of innovative materials to energy applications[11–20]. Especially, both theoretical and experimental explorations have suggested that isolated single metal-$N_x$ (M-$N_x$) modified carbon-based materials can serve as desirable oxygen electrocatalysts with promising performance[21–29]. Particularly, density functional theory (DFT) calculations demonstrate the standard symmetrical planar four-coordinated structure (denoted as M-$N_4$ moiety) might serve as the most favorable catalytic site for M-$N_x$ catalysts, seemingly supported by plenty of experimental results[30–33]. But some recent researches also point out that for the M-$N_4$ moiety, the large electronegativity of the symmetrical neighboring nitrogen atoms around the metal site would result in unsuitable free energy for adsorption the intermediate products[34,35]. Obviously, the non-optimal adsorption of the ORR intermediates badly decreases the kinetic activity and hampers the performance. As a solution to overcome the obstacles, the adsorption strength of ORR intermediates in the active sites could be modified by adjusting the interface configuration of the central metal atoms to reduce the potential barriers, which results in boosted catalytic activity[36,37]. Due to the comparative weak electronegativity, sulfur-permeating seems to be an attractive method to adjust the electronic structures of the active sites, realizing the improvement of ORR performance[38,39]. Conventionally, the alien sulfur atoms are anchored in the carbon matrix surrounded by C or N atoms, separated from the metal centers[40–43]. This regulation type of sulfur species can tune and enhance the kinetic activity of the M-$N_4$ site by adjusting electron-withdrawing/donating properties. However, in this situation, the activity modification by the doped sulfur is indirect and limited. What about the direct engagement of metal and sulfur atoms? It means that at least one nitrogen atom in the symmetrical M-$N_4$ moiety has to be kicked off by sulfur invaders. Will the adjacent pairs of metal and sulfur to construct an unsymmetrical atomic interface to produce boosted effects for ORR? As far as we know, few reports have addressed this question[44–46].

Herein, we developed a hierarchically porous carbon based single copper atom catalyst toward ORR, by rationally controlling the unsymmetrical interface structure of central metal atoms, in which Cu was directly bonded with both sulfur and nitrogen atoms (denoted as S-Cu-ISA/SNC). The engineered S-Cu-ISA/SNC demonstrated a half-wave potential of 0.918 V vs. RHE in alkaline media, which reflected its boosted ORR performance. The activity of S-Cu-ISA/SNC compared to related materials follows the trend: S-Cu-ISA/SNC (single-atom Cu-$S_1N_3$ supported on N and S co-doped carbon polyhedron) >Cu-ISA/SNC (single-atom Cu-$N_4$ supported on N and S co-doped carbon polyhedron) >Cu-ISA/NC (single-atom Cu-$N_4$ supported on N doped carbon polyhedron) >Pt/C. Moreover, S-Cu-ISA/SNC displayed excellent stability with no obvious current decay after long-term ORR test. X-ray absorption near-edge structure (XANES) and extended X-ray absorption fine structure (EXAFS) revealed that the outstanding ORR activity originated from the formation of the unsymmetrical Cu-$S_1N_3$ atomic interface in the carbon matrix, and we also discovered that low-valent Cu (+1) species worked as active sites for ORR. Furthermore, this strategy of atomic interface engineering could be used to other metals (Mn, Fe, Co, Ni, etc.).

## Results

**Synthesis and morphology characterizations of S-Cu-ISA/SNC.** The sample was prepared through a three-step process (Supplementary Fig. 1). In step one, zeolitic imidazolate frameworks (ZIF-8) were adopted as molecular-scale cages to absorb and encapsulate the copper precursor. Typically, Cu(acac)$_2$ was mixed with the precursors of ZIF-8 ($Zn^{2+}$ and 2-methylimidazole), and through a self-assembly process, Cu(acac)$_2$ were committed to the ZIF-8 cages (Cu-ZIF-8). In the second step, Cu-ZIF-8 and sulfur powder were jointly dispersed in carbon tetrachloride ($CCl_4$) and then dried by stirring, ensuring the sulfur was absorbed on the surface of Cu-ZIF-8 powder (labeled as S-Cu-ZIF-8, Supplementary Figs. 2 and 3). In the final step, S-Cu-ISA/SNC was obtained after the pyrolyzation of the S-Cu-ZIF-8 at 950 °C under Ar atmosphere. It was necessary to noted that the formed metallic zinc was evaporated (>907 °C) and meanwhile sulfur permeated in the ZIF-8 frameworks during pyrolysis[47,48]. Cu-ISA/SNC (S was separated from Cu), Cu-ISA/NC (S free), SNC (S, N co-modified carbon) and NC (N-modified carbon) were also prepared as comparison.

The synthetic samples were characterized by Powder X-ray diffraction (PXRD) patterns and Raman spectra (Supplementary Fig. 4). The results indicated that the ZIF-8 derived carbon frameworks were poorly crystallized after pyrolysis and also implied that plenty of defects existed in the carbon substrate, which was favorable for the anchoring of isolated metal atoms[49]. The morphology of S-Cu-ISA/SNC was observed by scanning electron microscopy (SEM) and transmission electron microscopy (TEM). Figure 1a showed that S-Cu-ISA/SNC roughly remained the polyhedral shape, but the surfaces became extremely bumpy. The TEM images (Fig. 1b and Supplementary Fig. 5a) indicated that the obtained sample possessed a highly open porous structure, meanwhile small Cu particles were not detected. The high-resolution transmission electron microscopy image (HRTEM) in Supplementary Fig. 5b told us that graphite carbon layers existed in the porous frameworks, which were beneficial for promoting the conductivity[50]. $N_2$ adsorption-desorption isotherms (Supplementary Fig. 6) demonstrated the fairly high specific surface area and the hierarchically porous characteristics of S-Cu-ISA/SNC. Our further in-situ environmental microscopic studies (Supplementary Figs. 7, 8, Supplementary Note. 1 and Supplementary Movies 1-2) suggested that the permeation of sulfur played an important role for etching the carbon frameworks. The hierarchically porous architecture could facilitate the charge and mass transportation for electrochemical reactions[51]. Energy-dispersive X-ray spectroscopy (EDS) (Fig. 1c and Supplementary Fig. 9) in the scanning transmission electron microscope (STEM) indicated Cu, S and N on the support were distributed uniformly. The Cu content in S-Cu-ISA/SNC was 0.73 at%, according to the ICP-OES results. The monodispersion of Cu could be directly monitored by spherical aberration STEM (Fig. 1d, e and Supplementary Fig. 10). The Cu atoms were confirmed by isolated bright dots in the high-magnification HAADF-STEM image. The sizes of dots were below 2.0 Å as shown in Supplementary Fig. 11. As elucidated in Fig. 1f, the distance between Cu atoms was more than 0.38 nm, which

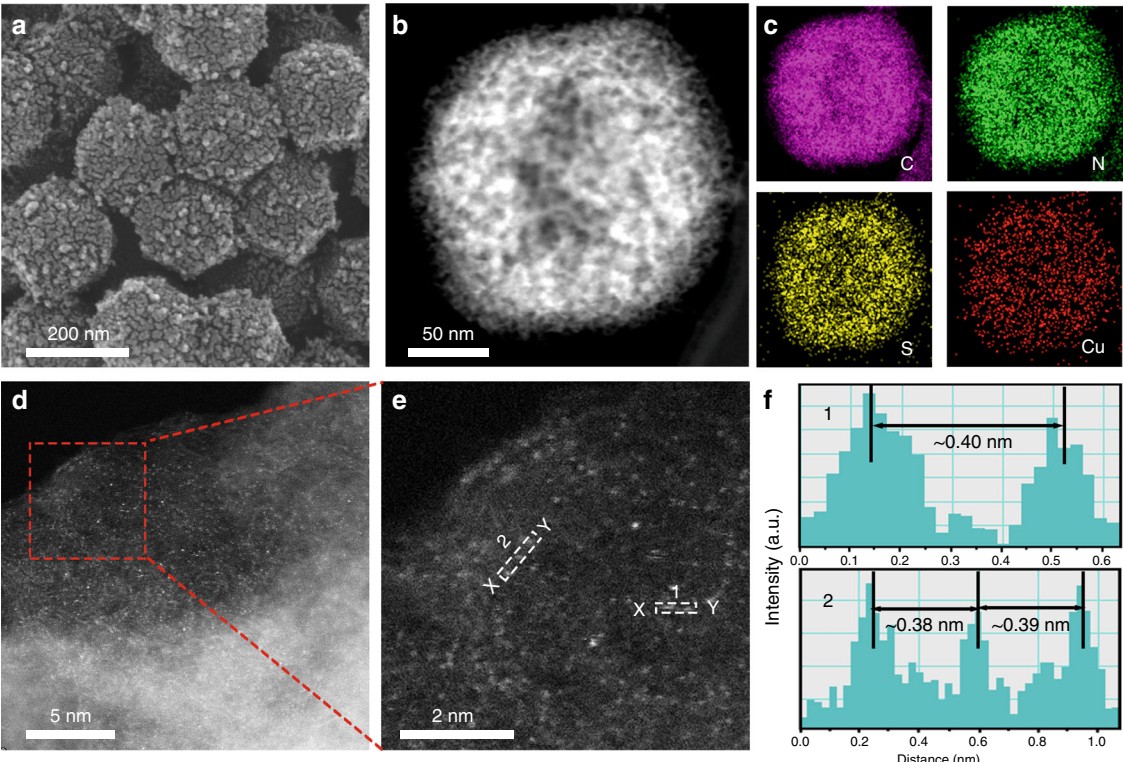

**Fig. 1 Morphology and composition characterizations of S-Cu-ISA/SNC. a** SEM, **b** TEM and **c** EDS images of S-Cu-ISA/SNC, C (pink), N (green), S (yellow) and Cu (red). **d** HAADF-STEM image and **e** the magnified image of S-Cu-ISA/SNC. **f** The corresponding intensity profiles along the line X-Y in **e**.

confirmed that Cu existed in isolated feature in S-Cu-ISA/SNC. Furthermore, the SEM, TEM and HAADF-STEM characterizations of NC, SNC, Cu-ISA/NC and Cu-ISA/SNC were also exhibited, respectively (Supplementary Figs. 12–15). We found that all the S-added samples (S-Cu-ISA/SNC, Cu-ISA/SNC and SNC) displayed etched porous feature, compared to those without sulfur participation (Cu-ISA/NC and NC). Additionally, the Zn content in S-Cu-ISA/SNC was as low as 0.028 at% according to the ICP-OES analysis, which excluded the possible influence to catalytic performance by the residue Zn species[52].

**Chemical state and atomic structure analysis of S-Cu-ISA/SNC.**
To probe the electronic and atomic interplay of Cu, S, N and C in S-Cu-ISA/SNC, synchrotron-radiation-based soft XANES was carried out (Supplementary Note. 2)[53]. The $L_3$ edge and $L_2$ edge of Cu XANES in S-Cu-ISA/SNC located at 931.2 eV and 950.9 eV (Fig. 2a). The L-edge position of S-Cu-ISA/SNC was between those of CuPc and CuS, implying the possible formation of Cu-S and Cu-N bonds, which was consistent with the XPS results (Supplementary Fig. 16). The carbon K-edge spectrum of S-Cu-ISA/SNC (Fig. 2b) was dominated by four clearly peaks located at 285.5 eV (peak a), 287.4 eV (peak $b_1$), 288.5 eV (peak $b_2$) and 292.4 eV (peak c), which could be attributed to the dipole transition of the C 1 s core electron to the $\pi^*C = C$, $\pi^*C-N/S-C$, and $\sigma^*C-C$ orbitals[54]. The peak $b_1$ and peak $b_2$ suggested the existing of Cu–N/S bonds at carbon matrix[55]. In addition, the electronic state of N in S-Cu-ISA/SNC could also be detected by the N K-edge XANES spectrum (Fig. 2c). The peaks $e_1$, $e_2$ and f indicated the pyridinic and pyrrolic nitrogen; the peak g denoted graphitic nitrogen[54]. The Cu-N bond was also monitored by N 1s XPS spectrum (Supplementary Fig. 16e). Furthermore, the S L-edge XANES spectrum (Supplementary Fig. 17) of S-Cu-ISA/SNC showed obvious peaks (peak h-j) in the region of 163–167 eV corresponding to C-S-C coordination species, suggesting the

anchor of S in the carbon skeleton[56]. The sulfur was further investigated by S K-edge XANES (Supplementary Fig. 18). In general, the valence of S was linear correlated to the K-edge position. We found that the sulfur in S-Cu-ISA/SNC was slightly positive charge, which might be attributed to the existence of S-N coordination, since N had higher electronegativity than S, as well as the existence of C-$SO_x$ species in the sample. The S K-edge EXAFS for S-Cu-ISA/SNC demonstrated the presence of S-C/N and S-Cu bonding, with FT peaks located at 1.3 Å and 2.1 Å, respectively (Supplementary Fig. 19).

X-ray absorption fine structure (XAFS) was carried out to gain insight into the interface structure at atomic scale. The position of the Cu K-edge absorption threshold was the reflection of average oxidation state of Cu species[57,58]. As illustrated in Fig. 2d, the edge position of S-Cu-ISA/SNC was between CuS and CuPc, demonstrating the average oxidation state of Cu was between the two references. In supplementary Fig. 20, the fitted oxidation state of Cu in S-Cu-ISA/SNC from K-edge XANES spectra was 1.97, agreeing well with XPS and soft L-edge XANES analysis. The Fourier transform (FT) EXAFS spectra of S-Cu-ISA/SNC and the references (Cu foil, CuS and CuPc) were illustrated in Fig. 2e. We found that the sample exhibited one obvious FT peak located at 1.55 Å, which was mainly attributed to the scattering of Cu-N coordination. Surprisingly, a shoulder peak located at 1.81 Å was also detected. By contrast with other FT-EXAFS spectra, this signal in S-Cu-ISA/SNC was considered owing to Cu-S scattering (Supplementary Figs. 21 and 22), which indicated the formation of Cu-S bonding. Furthermore, there was no related peak corresponding to Cu-Cu coordination, compared with Cu foil. Due to the powerful resolution in both k and R spaces, the Cu K-edge wavelet transform (WT)-EXAFS was applied to investigated the atomic configuration of S-Cu-ISA/SNC (Fig. 2f)[59]. By comprehensive consideration of the Cu-N and Cu-S contributions, the WT contour plots in S-Cu-ISA/SNC exhibited the

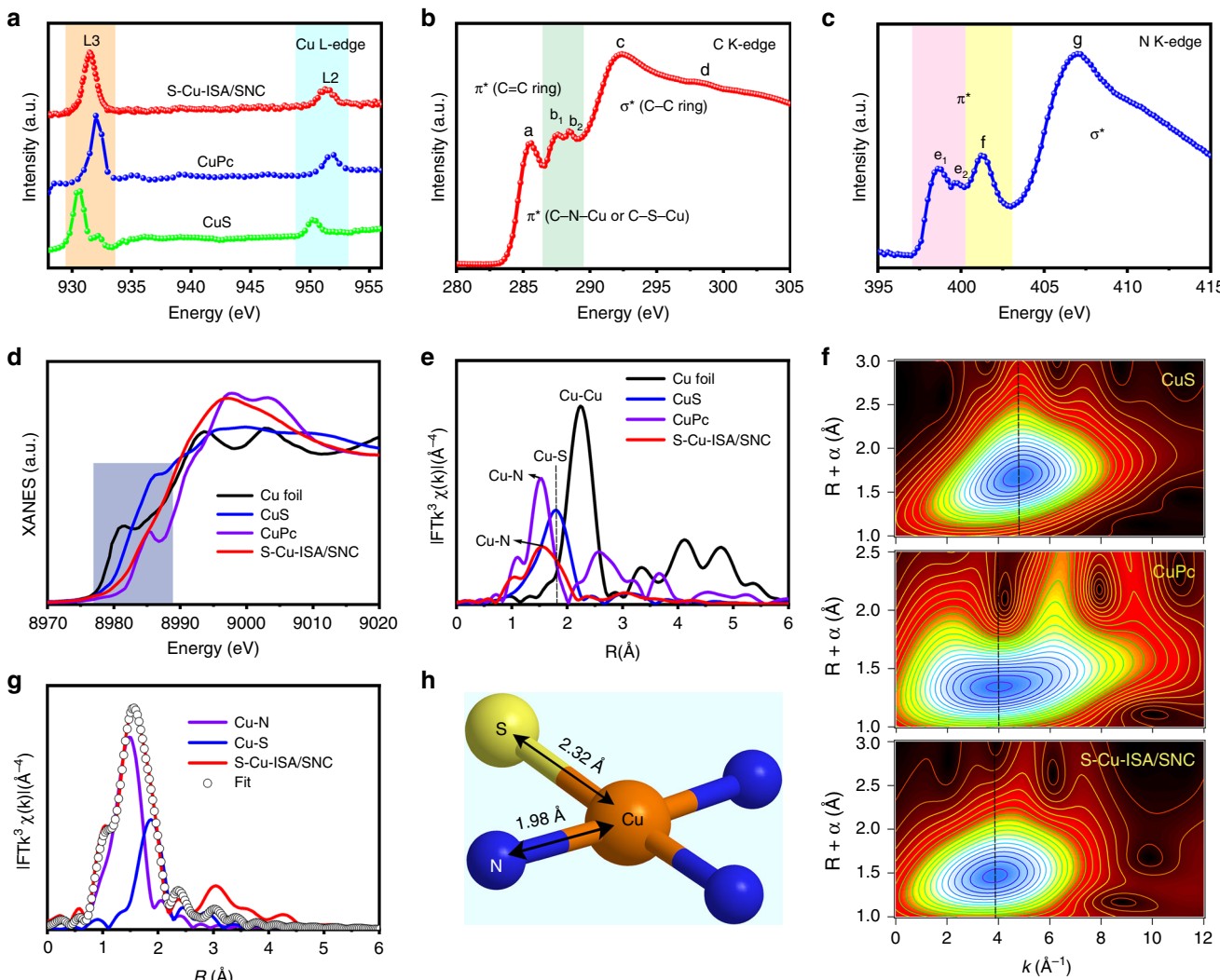

**Fig. 2 Chemical state and atomic local structure of S-Cu-ISA/SNC. a** Cu L-edge XANES spectra of S-Cu-ISA/SNC, CuS and CuPc. **b** C K-edge and **c** N K-edge XANES spectra of the S-Cu-ISA/SNC. **d** The experimental Cu K-edge XANES spectra of S-Cu-ISA/SNC and the references (Cu foil, CuS and CuPc). **e** FT $k^3$-weighted Cu K-edge EXAFS spectra of S-Cu-ISA/SNC and the references. **f** WT-EXAFS plots of S-Cu-ISA/SNC, CuS and CuPc, respectively. **g** FT-EXAFS fitting curves of S-Cu-ISA/SNC at Cu K-edge. **h** Schematic atomic interface model of S-Cu-ISA/SNC.

maximum peak at 3.9 Å$^{-1}$. In addition, compared with the WT signals of Cu foil, no Cu-Cu coordination was observed in S-Cu-ISA/SNC (Supplementary Fig. 23). These further identified the isolated feature of Cu species in S-Cu-ISA/SNC[60].

Quantitatively, the structural parameters at Cu K-edge was extracted by least-square EXAFS fitting. The results were exhibited in Fig. 2g, Supplementary Fig. 24 and Supplementary Table 1. It was observed that the fitting curves matched quite well with the experiment spectra. Depend on the results, the first shell of the central atom Cu displayed a coordination number of four, directly connected by one S atom and three N atoms, with the mean bond lengths of 2.32 Å and 1.98 Å, respectively (Fig. 2h). Furthermore, we investigated the simulated EXAFS spectra based on the models of Cu-S$_1$N$_3$, Cu-S$_2$N$_2$, Cu-S$_3$N$_1$ and Cu-N$_4$, given in Supplementary Fig. 25a. We could find that when the atom number of sulfur increased from one to three, the FT peak intensity of Cu-S increased understandably, compared to that of Cu-N[61]. The relative intensity of Cu-S and Cu-N in the Cu-S$_1$N$_3$ curve accorded quite well with the experimental spectrum. The theoretical XANES spectrum was also calculated based on the Cu-S$_1$N$_3$ model (Supplementary Fig. 25b) as well as Cu-N$_4$, Cu-S$_2$N$_2$ and Cu-S$_3$N$_1$ (Supplementary Fig. 26). We could see that the

calculation result for Cu-S$_1$N$_3$ could best reproduce the main features of the experimental curve of S-Cu-ISA/SNC. Moreover, we also tried linear combination fitting (LCF) of the experimental spectrum with the calculated spectrum for Cu-N$_4$ and experimental spectra for CuS and/or Cu$_2$S, as shown in Supplementary Figs. 27–29. We found that although the fitted curves near the edge seemed coincide with the experimental spectrum in some way, the curves after the white line were quite different, suggesting the absence of copper sulfide species. Based on the EXAFS fittings and simulations together with XANES calculations, the unsymmetrical Cu-S$_1$N$_3$ moiety in S-Cu-ISA/SNC was appropriately confirmed. The EXAFS results of Cu foil, CuS and CuPc were also exhibited in Supplementary Fig. 30 and Supplementary Table 1. By contrast, the EXAFS analysis of Cu-ISA/NC and Cu-ISA/SNC were showed in Supplementary Figs. 31, 32 and Supplementary Table 1, respectively. Both the Cu species in Cu-ISA/NC and Cu-ISA/SNC existed in the form of symmetrical Cu-N$_4$, different from that of S-Cu-ISA/SNC.

**Electrocatalytic performance of S-Cu-ISA/SNC on ORR.** The ORR activity of S-Cu-ISA/SNC was then evaluated in a typical three-electrode system (Supplementary Figs. 33 and 34).

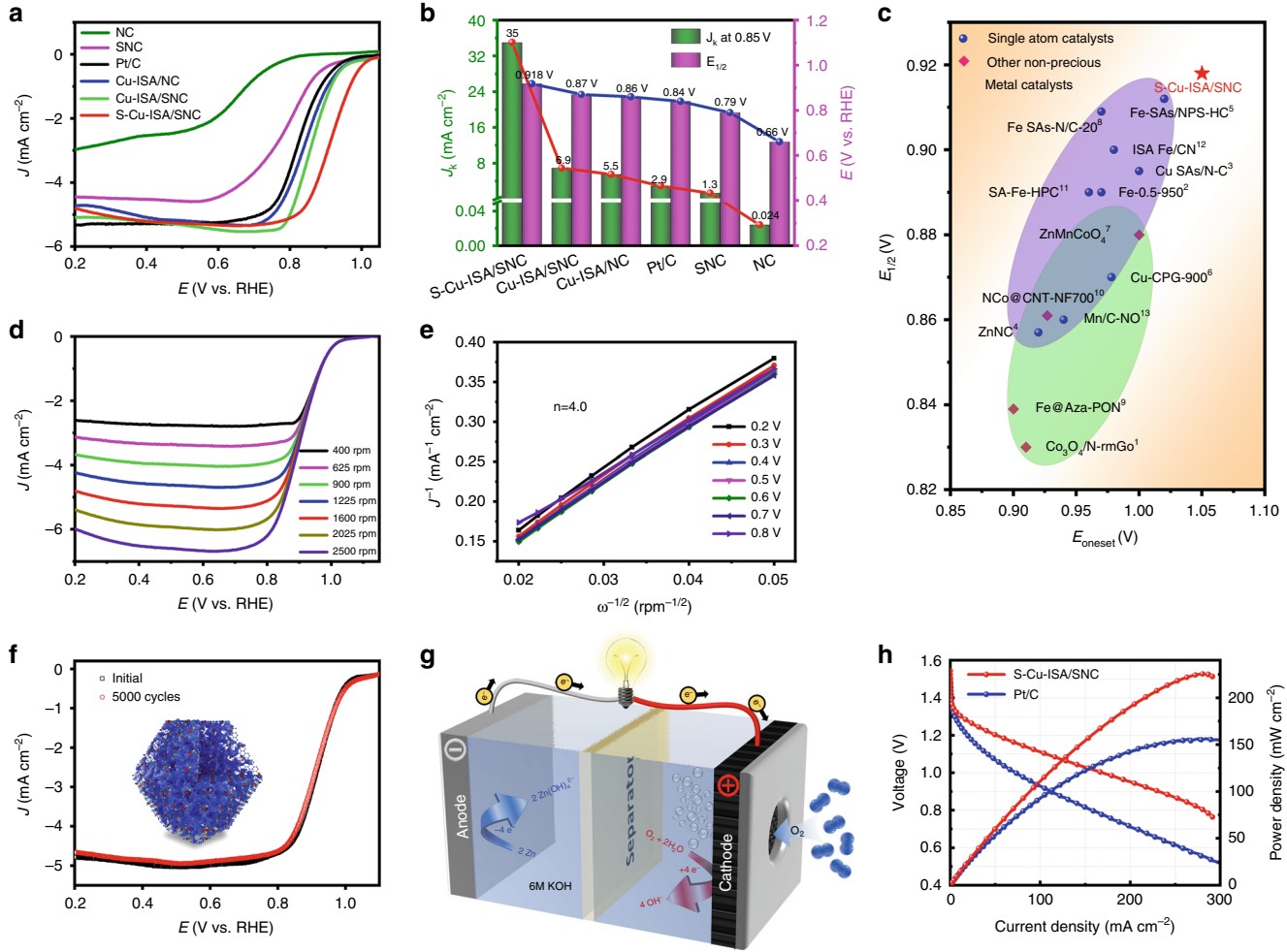

**Fig. 3 ORR activity of S-Cu-ISA/SNC. a** Polarization curves for S-Cu-ISA/SNC and the references. **b** The contrast between S-Cu-ISA/SNC and the references for $J_k$ (0.85 V) and $E_{1/2}$. **c** Contrasting the $E_{onset}$ and $E_{1/2}$ values for S-Cu-ISA/SNC and the catalysts in Supplementary Table 2. **d** The polarization curves of S-Cu-ISA/SNC at different rotating speeds. **e** The K-L plots for S-Cu-ISA/SNC. **f** The long-term durability tests of S-Cu-ISA/SNC, which was assessed by cycling the catalyst between 1.1 and 0.2 V vs. RHE at 50 mV s$^{-1}$. **g** Schematic diagram of Zn-air battery. **h** Discharge polarization curves and power density plots of S-Cu-ISA/SNC and Pt/C-based Zn-air batteries.

As illustrated in Fig. 3a, b and Supplementary Fig. 35 (the CV curves of S-Cu-ISA/SNC and Pt/C were exhibited in Supplementary Fig. 36), all the three single copper atom samples (S-Cu-ISA/SNC, Cu-ISA/SNC and Cu-ISA/NC) showed an optimistic performance. Especially, the S-Cu-ISA/SNC displayed an optimal activity with the highest kinetic current density ($J_k = 35$ mA cm$^{-2}$), as well as the most positive onset potential ($E_{onset}$) at 1.05 V and half-wave potential ($E_{1/2}$) at 0.918 V among the studied catalysts, and the catalytic activities of these catalysts followed the trend S-Cu-ISA/SNC > Cu-ISA/SNC > Cu-ISA/NC. The $E_{1/2}$ of S-Cu-ISA/SNC was even 78 mV higher than that of commercial Pt/C (0.84 V). By contrast, NC and SNC frameworks derived from ZIF-8 demonstrated rather low $J_k$ (0.024 mA cm$^{-2}$ and 1.3 mA cm$^{-2}$) and $E_{1/2}$ (0.66 V and 0.79 V, respectively), which indicated that in single copper atom catalysts, the Cu-S/N or Cu-N sites might serve as the active sites during ORR instead of the N-C or S-N-C. Furthermore, S-Cu-ISA/SNC surpassed all the other listed Cu-based ORR catalysts including some recently reported single Cu atom catalysts with isolated symmetrical Cu-N$_4$ centers[62,63]. The ORR activity of S-Cu-ISA/SNC was also compared with that of other nanostructured or single atom non-precious metal (Mn, Fe, Co, Ni, etc.) catalysts, and we found that S-Cu-ISA/SNC still demonstrated superior activity than those of them (Fig. 3c and Supplementary Table 2).

Koutecky-Levich plots of S-Cu-ISA/SNC were obtained from linear sweep voltammetry (LSV) curves (Fig. 3d). The calculated electron transfer number of S-Cu-ISA/SNC was 4.0 (Fig. 3e), which was the same as the theoretical value for Pt/C. As shown in the Supplementary Fig. 37, from 0.2 to 0.9 V, the electron transfer number for S-Cu-ISA/SNC was in the range of 3.92-3.99 and the H$_2$O$_2$ yield remained below 4%, indicating that the catalytic process on the S-Cu-ISA/SNC electrode underwent a high efficiency four-electron ORR process. The Tafel slope for S-Cu-ISA/SNC (50 mV decade$^{-1}$) was much lower than that of Pt/C (90 mV decade$^{-1}$), further conforming the excellent ORR activity for S-Cu-ISA/SNC (Supplementary Fig. 38). Supplementary Figs. 39, 40 demonstrated that S-Cu-ISA/SNC exhibited excellent methanol tolerance. In Fig. 3f, after 5000 cycles, little change in $E_{1/2}$ was observed for S-Cu-ISA/SNC. The chronoamperometry at 0.90 V vs. RHE of S-Cu-ISA/SNC catalyst showed that the ORR current remained 98% after 100 h test (Supplementary Fig. 41). The HAADF images and EXAFS spectra (Supplementary Figs. 42, 43) also proved that S-Cu-ISA/SNC had excellent stability for ORR. When tested in acidic media (0.5 M H$_2$SO$_4$ solution), the S-Cu-ISA/SNC catalyst also exhibited improved activity (Supplementary Fig. 44). The catalyst displayed $E_{1/2}$ of 0.74 V. The Tafel slope was 106.9 mV decade$^{-1}$. Furthermore, it showed comparable activity compared with other catalyst shown in

Supplementary Table 3. In addition, S-Cu-ISA/SNC in acid possessed good stability as well (Supplementary Fig. 44f).

Additionally, we tested the potential application of S-Cu-ISA/SNC in a home-made Zn-air battery (Fig. 3g and Supplementary Fig. 45a). As exhibited in Fig. 3h, the Zn-air battery using S-Cu-ISA/SNC catalyst as the air cathode displayed good activity. The maximum power density was 225 mW cm$^{-2}$, outperformed Pt/C (155 mW cm$^{-2}$) as well as the listed catalysts in Supplementary Table 4. In Supplementary Fig. 45b, the specific capacity of the battery employing S-Cu-ISA/SNC as air-cathode was estimated to be 735 mAh g$^{-1}$ at the discharge of 10 mA cm$^{-2}$. Moreover, the S-Cu-ISA/SNC-based battery could robustly serve up to 50 h with little discharge voltage decrease (Supplementary Fig. 45c), which indicated the outstanding durability for S-Cu-ISA/SNC based device.

**In situ XAS measurements of S-Cu-ISA/SNC.** In order to monitor the structural evolution of the isolated copper sites during ORR, potential-dependent Cu K-edge XAS of S-Cu-ISA/SNC was carried out[64,65]. The in situ XAS tests were carried out using a home-made cell (Fig. 4a and Supplementary Fig. 46), and all the spectra were collected in fluorescence model by a common-used Lytle detector. The S-Cu-ISA/SNC sample was uniformly dropped on a carbon paper, ensuring that all the Cu species took part in the ORR reaction (Supplementary Fig. 47). Firstly, the possible X-ray radiation damage on S-Cu-ISA/SNC was examined (Supplementary Fig. 48a), and it was found that the XANES region at Cu K-edge was with no obvious change after a longtime irradiation (2 h), suggesting that the radiation damage was negligible. Then the sample-loaded carbon paper was immersed in 0.1 M KOH solution, without electricity and oxygen inpouring. The collected XANES spectra (Supplementary Fig. 48b) implied that the solution has little influence on the

structure before ORR test. The Cu K-edge in situ XANES spectra for S-Cu-ISA/SNC was examined at different potentials (Supplementary Fig. 49). The results were displayed in Fig. 4b and Supplementary Fig. 50, respectively. From 1.05 V to 0.75 V, the edge position was gradually moved to the lower energy, together with reduce of the white line intensity, which suggested a decrease of the valence of Cu in S-Cu-ISA/SNC during ORR. The trend could be reflected more clearly from the XANES difference curves (Fig. 4c). The average oxidation states (Fig. 4d and Supplementary Figs. 51 and 52) indicated that the valence of Cu species decreased from approximately +2 to +1, implying that Cu (+1) sites might work as the active centers for ORR[66,67]. When the applied potential returned from 0.75 V to 1.05 V, Cu XANES edge shifted back to higher energy along with increase of the white line peak (Supplementary Figs. 53 and 54). This provided unequivocal evidence that the XANES spectra as a function of applied potential were reversible, which might be due to the strong anchor effect of N and S atom to the Cu sites. The reversible change of Cu valence state was a reflection of its significant contribution to the outstanding catalytic activity for ORR.

In addition, in situ EXAFS was conducted to monitor the atomic interface structure of the Cu sites during ORR (Fig. 4e and Supplementary Fig. 55). Figure 4e showed the corresponding $k^3$-weighted FT-EXAFS spectra for S-Cu-ISA/SNC at 0.90 V and 0.75 V vs. RHE. Just like the ex situ data, the in situ FT-EXAFS curves still exhibited one main peak (Cu-N) along with a shoulder peak (Cu-S). However, under the realistic condition, the Cu-N peaks appeared an obvious low-R move from 1.55 Å to 1.49 Å. This implied that the local structure of the active site was changed, which was monitored through the shrinking of Cu-N bond length. The EXAFS curve-fitting results were exhibited in Supplementary Figs. 56, 57 and Supplementary Table 5, where

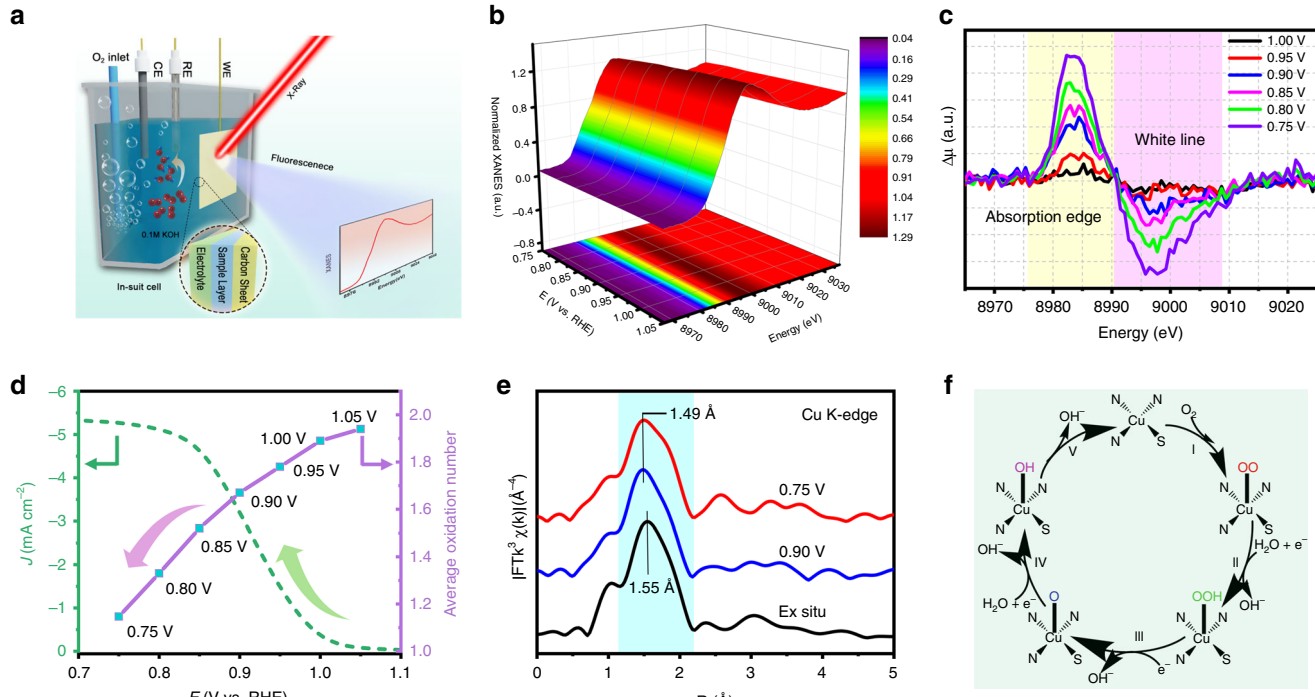

**Fig. 4 In situ XAFS characterization of S-Cu-ISA/SNC. a** Schematic of the in situ electrochemical cell set-up. CE, counter electrode; WE, working electrode; RE, reference electrode. **b** Cu K-edge XANES spectra of S-Cu-ISA/SNC at various potentials during ORR catalysis in O$_2$-saturated 0.1 M KOH. **c** Differential Δμ XANES spectra obtained by subtracting the normalized spectrum at every potential to the spectrum recorded at 1.05 V vs. RHE. **d** Current density as a function of potential for S-Cu-ISA/SNC (left) and the average oxidation number of Cu species in S-Cu-ISA/SNC as a function of potential (right). **e** $k^3$-weighted FT-EXAFS at ex-situ, 0.90 V and 0.75 V vs. RHE. The shaded region highlighted the variations in the peak position of the first coordination shell. **f** The proposed ORR mechanism for the S-Cu-ISA/SNC.

three backscattering paths including Cu-N, Cu-O and Cu-S were considered. The ex situ spectrum indicated the Cu-N bond lengths of 1.98 Å, while the bond lengths were shortened to 1.94 Å (0.90 V vs. RHE) and 1.93 Å (0.75 V vs. RHE), respectively, at real-time working conditions. The most possible geometric configuration was considered as an isolated unsymmetrical Cu-$S_1N_3$ moiety linked with the OOH*, O* and OH* intermediates as shown in Fig. 4f, which was also detected by our in situ Fourier Transform infrared spectroscopy (FTIR) test (Supplementary Fig. 58)[68]. At the same time, the Cu-S bond lengths were detected to be nearly unchanged and kept at about 2.32 Å. Moreover, Supplementary Fig. 59 showed the HAADF-STEM images at different areas of the S-Cu-ISA/SNC catalyst after in situ XAS tests, which suggested the absence of Cu clusters or small copper sulfide species. In short, the in situ spectroscopy analysis elucidated the electronic and atomic structure evolution of the Cu-$S_1N_3$ moiety in S-Cu-ISA/SNC and revealed that the low-valence (+1) Cu-N-bond-shrinking HOO-Cu-$S_1N_3$, O-Cu-$S_1N_3$ and HO-Cu-$S_1N_3$ species might contribute to the good ORR activity.

**Theoretical study of S-Cu-ISA/SNC on ORR.** To understand the observed enhancement of ORR activity for S-Cu-ISA/SNC, DFT calculations were conducted to analyze the whole process of the four-electron ORR reaction on different Cu-centered moieties embedding in carbon matrix. Considering that the atomic radius of S was much larger than that of N or C atoms (Supplementary Table 6), we substituted two adjacent atoms of C or N by the S atom to maintain the stability of sulfur-doped structures in our calculations (Supplementary Fig. 60 and Supplementary Note. 3). Meanwhile, as shown in Supplementary Table 7, the formation energy of S-doped moiety rapidly increased with the number of the coordinated sulfur atoms bonded with central Cu atom, suggesting that the moiety with multi-S coordinated atoms bonded with central Cu atom (symmetrical Cu-para-$S_2N_2$ or unsymmetrical Cu-ortho-$S_2N_2$) was much less stable than unsymmetrical Cu-$S_1N_3$. With this prediction of stability, we comparably investigated the ORR activities of S-Cu-ISA/SNC (S was coordinated with Cu), Cu-ISA/SNC (S was separated from Cu) and Cu-ISA/NC (S free), respectively, including the pristine graphene structures embedding with unsymmetrical Cu-$S_1N_3$ for S-Cu-ISA/SNC, S-doped graphene structures embedding with Cu-$N_4$ moieties (Cu-$N_4$-$S_1$-1 and Cu-$N_4$-$S_1$-2) for Cu-ISA/SNC, and the pristine graphene structures embedding with the Cu-$N_4$ moiety for Cu-ISA/NC (Fig. 5a, b, Supplementary Figs. 61 and 62 and Supplementary Tables 7 and 8). According to the Sabatier principle, the best catalysts which located at the vertex of volcano-type plot should bind reaction intermediates neither too strongly nor too weakly[69]. In ORR reaction, for catalysts (such as Fe-$N_4$ and Mn-$N_4$) that strongly binded intermediates, locating at the left side of volcano-type plot, the potential-limiting step was the desorption of OH* intermediate. While for catalysts (such as Co-$N_4$ and Ni-$N_4$) that weakly binded intermediates, locating at the right side of volcano-type plot, the potential-limiting step was the adsorption of OOH* intermediate[70]. Fig. 5a showed that the ORR activity of Cu-$N_4$ was far away from the vertex of the volcano-type plot and locates at the right side, suggesting that the Cu atom in Cu-$N_4$ moiety binded ORR intermediates too weakly[34,71]. With the introduction of sulfur atoms, the ORR activities were improved greatly. Particularly, the Cu atom in unsymmetrical Cu-$S_1N_3$ moiety had the best ORR activity among all Cu-centered moieties (Fig. 5a, b), with the overpotential of 0.39 V, which was even better than that of Fe-$N_4$ moiety. Thus, we demonstrated that the formation of the unsymmetrical Cu-$S_1N_3$ atomic interface in the carbon matrix benefited the improved ORR activity of the catalyst, which was consistent with the experimental results.

To further investigate the physical origin of the superior ORR performance for S-Cu-ISA/SNC, we also analyzed the electronic structures feature of different Cu-center moieties. As the electronegativity of S was smaller than that of N (Supplementary Table 6), Cu in S-Cu-ISA/SNC was likely to lose less valence electron since one coordinated N was substituted by S than Cu in Cu-ISA/NC (Fig. 5c and Supplementary Table 8)[72]. However, as shown in Fig. 5c, there was no clear linear correlation between the number of Bader charge of Cu and the adsorption free energy of O* for different moieties, suggesting that the superior ORR performance of S-Cu-ISA/SNC was not directly determined by the number of valance electron of Cu atom. Figure 5d, e showed the projected density of states (PDOS) for d orbitals of Cu before and after O* adsorption on the Cu-$S_1N_3$ atomic interface of S-Cu-ISA/SNC, respectively. Clearly, due to the introduction of the coordinated S, the Cu atom in the Cu-$S_1N_3$ moiety had more electrons which occupied the $d_{x^2-y^2}$ orbital than that in the Cu-$N_4$ moiety (Supplementary Fig. 63). After the O* adsorption, for the Cu-$N_4$ moiety, the p orbital of O and the $d_{z^2}$ orbital of Cu formed σ bond. Meanwhile, the p orbital of O and only $d_{yz}$ and $d_{xz}$ orbitals of Cu could form π bonds. For Cu-$S_1N_3$ moiety, the σ bond was also derived from the O p orbital and the Cu $d_{z^2}$ orbital, while the π bonds originated from the O p orbital and the Cu $d_{yz}$, $d_{xz}$ as well as $d_{x^2-y^2}$ orbitals, which was quite different from that of Cu-$N_4$ (Fig. 5f, Supplementary Fig. 63). Dramatically, the additional π bonds contributed from the Cu $d_{x^2-y^2}$ orbitals strengthened the weak bonding of ORR intermediates, resulting in the boosted ORR performance of Cu centers. Furthermore, it was clearly shown in Supplementary Fig. 61, for the unsymmetrical Cu-$S_1N_3$ atomic interface, the O* intermediates of ORR were not located exactly at the top site of Cu atom, so the O p orbital could interact with the Cu $d_{x^2-y^2}$ orbitals. While for the symmetrical Cu-$N_4$ moiety, the O* intermediates of ORR were located at the top site of Cu due to the symmetry confinement, without the interaction of O p orbital and Cu $d_{x^2-y^2}$ orbitals. Based on the experimental and theoretical results, the activity trend of ORR was well-confirmed.

**Synthesis and ORR performance of S-M-ISA/SNC (M = Mn, Fe, Co, Ni).** The synthetic method could expand to other 3d metal (Mn, Fe, Co and Ni, etc.) (Supplementary Table 9). HAADF-STEM images identified the isolated feature of Mn, Fe, Co and Ni in the obtained catalysts, which was further revealed by FT-EXAFS curves (Fig. 6 and Supplementary Figs. 64–71). Quantitative EXAFS fittings were also carried out (Supplementary Table 10), which suggested the center metal coordinated directly with N and S atom to form M-$S_1N_3$ moiety at the atomic surface. The extended study identified the universal of the synthetic strategy.

The ORR catalytic activities of S-M-ISA/SNC (Mn, Fe, Co, Ni) was then evaluated by electrochemical measurements in 0.1 M KOH. Supplementary Fig. 72 exhibited the LSV curves for S-Mn-ISA/SNC, S-Fe-ISA/SNC, S-Co-ISA/SNC and S-Ni-ISA/SNC. As we could see, the samples of S-M-ISA/SNC (M = Mn, Fe, Co, Ni) showed optimistic performance. The half-wave potential ($E_{1/2}$) of S-Mn-ISA/SNC, S-Fe-ISA/SNC, S-Co-ISA/SNC and S-Ni-ISA/SNC was 0.902 V, 0.917 V, 0.911 V and 0.851 V, respectively. The favorable ORR kinetics of S-Mn-ISA/SNC, S-Fe-ISA/SNC, S-Co-ISA/SNC and S-Ni-ISA/SNC was verified by kinetic current density ($J_k$) of 14.5, 40.0, 27.0 and 5.1 mA cm$^{-2}$ (Supplementary Fig. 73). The Tafel slope of S-Mn-ISA/SNC, S-Fe-ISA/SNC, S-Co-ISA/SNC and S-Ni-ISA/SNC was calculated to be 83.8, 62.6, 72.5 and 91.7 mV dec$^{-1}$ (Supplementary Fig. 74). These results further demonstrated the desirable ORR kinetics for S-M-ISA/SNC,

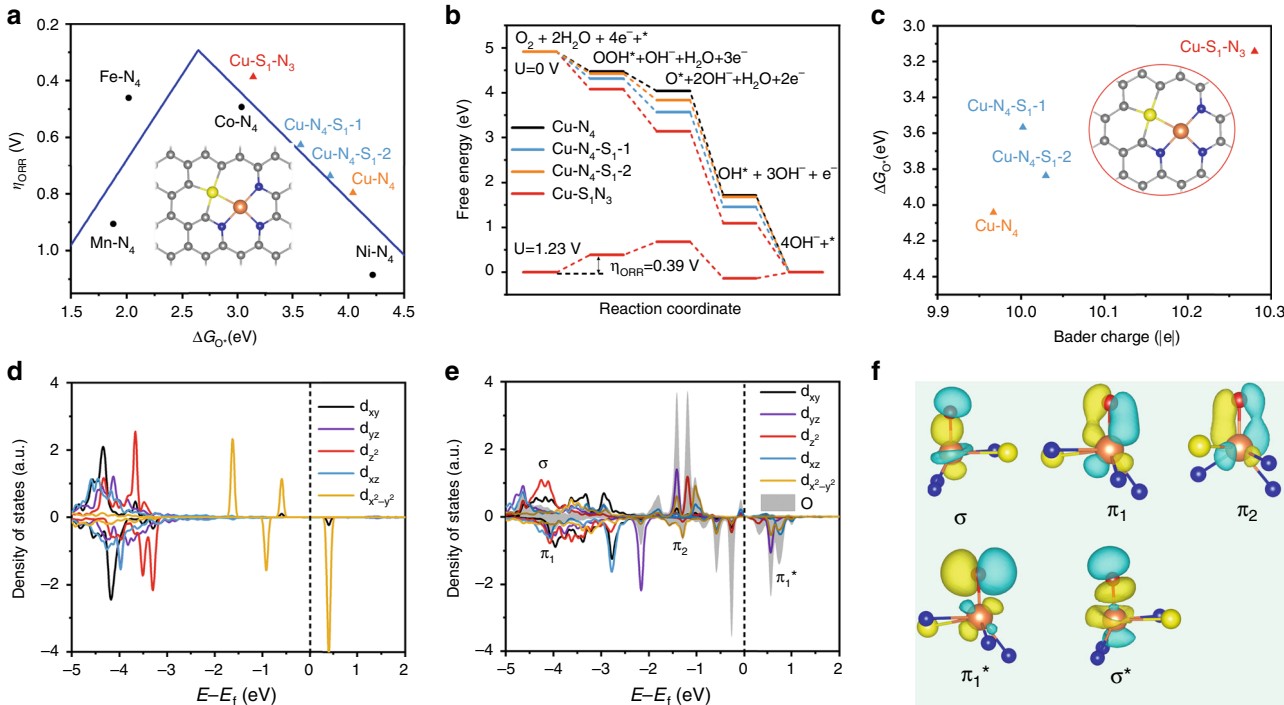

**Fig. 5 Theoretical ORR activity of S-Cu-ISA/SNC. a** ORR overpotential ($\eta_{ORR}$) as a function of O* adsorption free energy ($\Delta G_{O^*}$) on different Cu-centered moieties. Gray, blue, orange and yellow balls represent C, N, Cu and S atoms, respectively. **b** Free-energy diagram for different Cu-centered moieties. **c** Relationship between the number of Bader charge of Cu and $\Delta G_{O^*}$ for different Cu-centered moieties. Projected density of states of Cu and O* **d** before and **e** after O* adsorption for Cu-S$_1$N$_3$ in S-Cu-ISA/SNC. **f** Molecular orbitals of O* adsorbed on Cu-S$_1$N$_3$ in S-Cu-ISA/SNC. σ and σ* represent the bonding and antibonding between $d_{z^2}$ orbital of Cu and $p$ orbital of O, $\pi_1$ and $\pi_1^*$ represent the bonding and antibonding between $d_{yz}/d_{xz}$ orbital of Cu and $p$ orbital of O, $\pi_2$ represents the bonding between $d_{x^2-y^2}$ orbital of Cu and $p$ orbital of O.

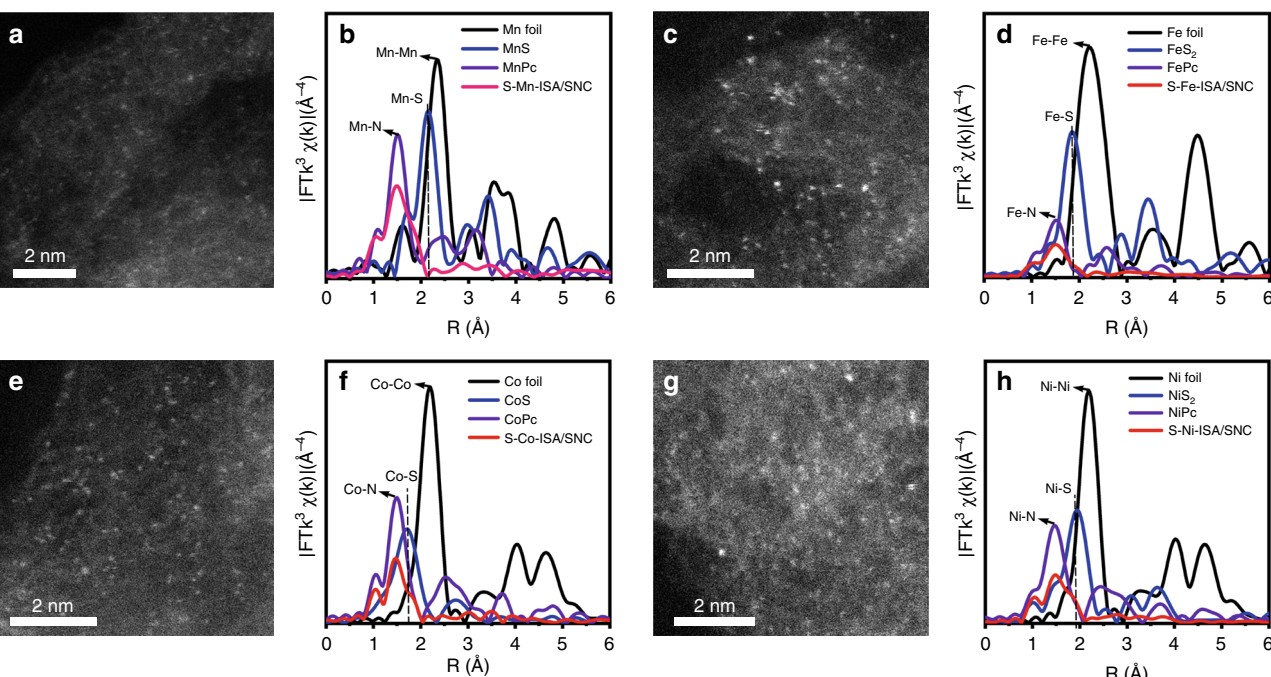

**Fig. 6 HAADF-STEM and FT-EXAFS characterization of S-M-ISA/SNC (M=Mn, Fe, Co, Ni).** HAADF-STEM images of **a** S-Mn-ISA/SNC, **c** S-Fe-ISA/SNC, **e** S-Co-ISA/SNC and **g** S-Ni-ISA/SNC. FT-EXAFS spectra of **b** S-Mn-ISA/SNC, **d** S-Fe-ISA/SNC, **f** S-Co-ISA/SNC and **h** S-Ni-ISA/SNC.

originating from the unsymmetrical M-$S_1N_3$ atomic interface structure.

## Discussion

In summary, we developed an single Cu atom ORR electrocatalyst consisting of unsymmetrical Cu-$S_1N_3$ complexes anchored in MOF-derived hierarchically porous carbon frameworks through an atomic interface engineering strategy. Benefiting from the rational construction of the active sites, the S-Cu-ISA/SNC sample exhibited outstanding ORR activity in alkaline media. Our experimental explorations and theoretical analysis revealed the enhanced ORR performance owed to the optimized atomic arrangement and density-of-states distribution of the Cu-$S_1N_3$ centers. The proposed strategy of local structure regulation may promote the research of advanced oxygen-involved reactions, as well as other electrochemical process.

## Methods

**Chemicals**. Cupric Acetate Monohydrate (Cu(acac)$_2$, 99%, Alfa Aesar), 2-methylimidazole (Acros), sulfur powder (325 mesh, 99.5%, Alfa), commercial Pt/C (20 wt% metal, Alfa Aesar), zinc nitrate hexahydrate (98%, Alfa Aesar), KOH (analytical grade, Sinopharm Chemical), carbon tetrachloride (innochem), analytical grade methanol (Sinopharm Chemical), Nafion D-521 dispersion (Alfa Aesar), N, N-dimethylformamide (DMF) (Sinopharm Chemical) were used without any further purification. The distilled water with a resistivity of 18.2 MΩ cm$^{-1}$ was used in all experiments.

**Preparation of S-Cu-ISA/SNC and the comparison samples**. In a typical synthesis of S-Cu-ISA/SNC catalyst, firstly the precursors were prepared by mixing sulfur powder and Cu-ZIF-8 (Supplementary Note. 4) in 20 ml of mixture solution (carbon tetrachloride: ethanol = 4:1) under sonication. The mass ration of sulfur powder and Cu-ZIF-8 is 1: 10. Subsequently, the solution was heated at 60 °C under vigorous stirring until drying. Afterwards, the samples were pyrolyzed in quartz tube. The pyrolysis process was in the Ar atmosphere, maintaining 450 °C (2 h) and then 950 °C (4 h). The ramping rate during the heating process was 5 °C/min. For the comparison samples, SNC (without the addition of Cu(acac)$_2$) was prepared as the same process. Cu-ISA/NC (single-atom Cu-N$_4$ supported on N doped carbon polyhedron, without the addition of S) and NC (N doped carbon polyhedron, without the addition of S and Cu) were obtained by pyrolysis of Cu-ZIF-8 and pure ZIF-8, respectively. The preparation of Cu-ISA/SNC (single-atom Cu-N$_4$ supported on N and S co-doped carbon polyhedron) was described in Supplementary Note. 5.

**Characterizations**. We used the SEM (JSM-6700F), TEM (JEOL-JEM-1200EX) and TEM (JEOR-2100F) to characterize the morphology. The in situ ETEM was carried out in Titan G2 60-300 microscope (FEI) equipped with a probe Cs-corrector, with voltage of 300 kV. Using JEOL JEM-ARM200F to gain the HAADF-STEM images, the accelerating voltage was 300 kV. The Bruker D8 ADVANCE X-ray Diffractometer was performed to characterize XRD patterns. HORIBA Jobin Yvon (LabRAM HR Evolution) was used to perform the Raman measurements with the laser of 532 nm. NOVA 4200e was used to obtain the BET surface area and the pore size distribution of the materials.

**Electrochemical measurements for ORR**. We used the three-electrode cell to perform the electrochemical tests. The working electrode was rotating disk electrode (glassy carbon), with a diameter of 5 mm. The counter electrode was graphite rod. The reference electrode was Ag/AgCl (filled with saturated KCl solution) electrode. The experiment was performed in 0.1 M KOH solution. All potentials have been converted to the RHE scale. Dispersing 1 mg catalyst to the mixture solution (0.75 ml isopropyl alcohol, 0.25 ml deionized water, 0.02 ml 5% Nafion), the catalyst ink was successfully prepared after sonication. The catalyst loading on the surface of the glassy carbon electrode was 0.102 mg cm$^{-2}$. Before ORR tests, we bubbled N$_2$/O$_2$ to make the system saturated. The CV tests of the catalyst under N$_2$- and O$_2$-saturated alkaline electrolyte were performed at 50 mV s$^{-1}$. LSV test was measured in different rotating rate from 400 to 2250 rpm. The electron selectivity was identified by rotating ring-disk electrode (RRDE) test (Supplementary Fig. 34). 1.23 V vs. RHE was applied as the ring electrode potential. At the same time, the disk electrode was performed at 10 mV s$^{-1}$. The detail for the electrochemical data processing was displayed in Supplementary Note. 6.

**Zinc-air battery measurements**. The S-Cu-ISA/SNC ink was uniformly dispersed onto teflon-coated carbon fiber paper, the loading is 1.0 mg cm$^{-2}$, then using 60 °C to make it dry. The Pt/C electrode was synthesized in the same way. Furthermore, the anode was commercial Zn foil (0.2 mm). And we polished it before use. The

Zn–air device was constructed by placing electrodes in O$_2$ saturated KOH solution (6M).

**Ex situ XAFS measurements**. The XAFS spectra (Cu, Mn, Fe, Co, Ni K-edge) were collected at 1W1B station in Beijing Synchrotron Radiation Facility (BSRF, operated at 2.5 GeV with a maximum current of 250 mA), BL14W1 station in Shanghai Synchrotron Radiation Facility (SSRF, 3.5 GeV, 250 mA) and BL7-3 station in Stanford Synchrotron Radiation Lightsource (SSRL, 3 GeV, ~500 mA), respectively. The XAFS data of the samples were collected at room temperature in fluorescence excitation mode using a Lytle detector. The samples were pelletized as disks of 13 mm diameter with 1 mm thickness using graphite powder as binder. The XAFS data processing was displayed in Supplementary Note. 7.

**In situ synchrotron radiation XAFS and FTIR measurements**. A catalyst modified carbon paper was used as working electrode, graphite rod as counter electrode and Ag/AgCl (KCl-saturated) electrode as reference electrode. A home-made electrochemical cell was used for in situ XAFS measurements (Fig. 4a and Supplementary Fig. 46). The experiments were performed at BL14W1 station in SSRF. The detail of in situ XAFS measurements was exhibited in Supplementary Note. 8. The in situ FTIR tests were performed at the BL01B at NSRL through a home-made set-up with a ZnSe crystal as the infrared transmission window. The detail for the in situ FTIR measurements is described in Supplementary Note. 9.

**The detail of DFT calculations**. Spin polarized DFT calculations were performed within the Vienna ab initio Simulation Package (VASP) with the projector augmented wave (PAW) scheme[73–75]. The exchange correlation energy was described by using the generalized gradient approximation (GGA) with the Perdew-Burke-Ernzerhof (PBE) functional[76] Hubbard corrected DFT (DFT + U) method was applied by considering on-site coulomb (U) and exchange (J) interaction, with the U-J values taken from the ones used by Xu et al.[33]. The cutoff energy was set to be 500 eV. The total energy and forces convergence thresholds were set to be 10$^{-5}$ eV and 0.02 eV Å$^{-1}$r, respectively. To prevent interaction between two neighboring surfaces, the vacuum layer thickness was set to 20 Å. The k-point sampling of the Brillouin zone was used by the 3 × 3 × 1 grid for structural relaxation and the 5 × 5 × 1 grid for electronic structure calculations. The empirical DFT-D3 correction was used to describe van der Waals (vdW) interactions[77]. Atomic charges were calculated by using the atom-in-molecule (AIM) scheme proposed by Bader[78,79]. Following the RHE model developed by Nørskov et al, the voltage-dependent ORR free energy pathway during electrocatalysis reaction were obtained[80]. The free energies of ORR intermediates are defined as $G = E_{DFT} + E_{ZPE} - TS$, where $E_{DFT}$, $E_{ZPE}$, T and S represent the calculated ground state energy, zero-point energy, temperature (298 K) and the entropy, respectively.

## Data availability

The data supporting the findings of this study are available within the article and its Supplementary Information files. All other relevant source data are available from the corresponding authors upon reasonable request.

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

## Acknowledgements

This work was supported by the National Key Research and Development Program of China (2017YFB0701600), the National Natural Science Foundation of China (Grant No. 51631001, 21801015, 51872030, 21643003, 51702016, 51501010, 11874036), the Guangdong Province Key Area R&D Program (2019B010940001), the Local Innovative and Research Teams Project of Guangdong Pearl River Talents Program (2017BT01N111), Basic Research Project of Shenzhen, China (JCYJ20170412171430026) and Beijing Institute of Technology Research Fund Program for Young Scholars (3090012221909), Fundamental Research Funds for the Central Universities. This work was also supported by the National Key R&D Program of China (2018YFA0702003) and the National Natural Science Foundation of China (21890383, 21671117, 21871159). The authors thank the BL1W1B and BL4B7A in the Beijing Synchrotron Radiation Facility (BSRF), BL14W1 in the Shanghai Synchrotron Radiation Facility (SSRF), BL01B, BL10B and BL12B in the National Synchrotron Radiation Laboratory (NSRL), BL11A in National Synchrotron Radiation Research Center (NSRRC), BL7-3 in Stanford Synchrotron Radiation Lightsource (SSRL) for help with characterizations. Tianjin and Guangzhou Supercomputing Center are also acknowledged for allowing the use of computational resources.

## Author contributions

W.C., D.W. and J.T.Z. conceived the idea, designed the study and wrote the paper. W.C. and H.S. carried out the sample synthesis, characterization and ORR measurement. W. C. carried out the XAFS characterizations and data analysis. J.D., Y.W., L.Z., R.C. and R.S. helped with the hard XAFS measurements and discussion. X.S.Z. helped with the XPS test. J.Z. and W.Y. helped with soft XAS test. A.L. performed the in-situ environmental microscopic measurements and analysis. Y.L. helped with the spherical aberration electron microscopy test and discussion. Z.Y. carried out the Zn-air battery measurements. Q.L. and X.Z. performed the in-situ IR test and analysis. X.Y.Z., X.Y. and J.L. performed the DFT calculations. J.P. and Z.Z. helped with the electrochemical tests and data analysis. Z.J., D.Z. T.S. and Z.L. helped with the modification of the paper. Y.D.L. gave very useful suggestions.

## Competing interests

The authors declare no competing interests.
