## [Peer Review File · Nature Communications]

Reviewers' Comments:

Reviewer #1:

Remarks to the Author:

In this work the authors present a CuN_xC_y catalyst containing sulphur atoms and showing an enhanced oxygen reduction reaction activity than the sulphur-free compounds. The morphology and active site structure are mainly investigated by electron microscopy techniques, X-ray absorption spectroscopy and DFT calculations. This is an interesting system, but there are a number of puzzling issues:

- 1) We know that X-ray absorption spectroscopy is a bulk technique, so how the authors can exclude the formation of copper sulphide species coexisting with CuN_x sites, rather than a new compound with formula CuN_xS_y ? This doubt is also fostered by the fact that the Cu-S bond length does not change under working conditions, and if CuS clusters are very small, they cannot be detected by TEM.
- 2) For the same reason of the previous point, authors cannot claim that a Cu^{+1} is formed and it is the active centre. We know that under applied potential CuNC forms very small Cu nanoparticles (DOI:10.1038/s41467-018-02819-7; DOI: <https://doi.org/10.1002/anie.201907994>, even if this is the extreme case of very low potential applied). This means that if a small fraction of Cu^{+2} is reduced to metallic copper, the average oxidation state will be lower than 2+. It is important to stress that EXAFS is not able to highlight a Cu-Cu contribution if the metallic fraction is small. Still, a peak around 2.5 Å appears in the FT-EXAFS spectrum at 0.75 V, and it is not necessary to find a peak exactly at same position of metallic copper to have a metallic-based cluster with maybe a different nature than that of crystalline copper.
- 3) Therefore, on the basis of the issue 1) and 2) the DFT model proposed by the authors may not be valid.

Other minor issues:

- 4) Why for the estimation of the oxidation number of the catalyst, only Cu foil and CuO references are used (figure 16 of supplementary materials)? This kind of determination requires attention, because the oxidation state (related to the XAS edge position) is not absolute, and it depends on the specific compound.
- 5) Why XAS spectra are just recorded between 0.75 and 1.05 V?
- 6) Is the in-situ XAS spectrum reversible when you come back to the OCV potential?
- 7) I personally do not appreciate images like Figure 1e with red circles around hypothetical sites. In many parts of these pictures the spots overlap and the hypothetical single-atoms site is not clear.
- 8) The authors completely forget to mention a lot of literature from other worldwide research groups, and not only from the restricted geographical area of most of the authors in the bibliography of this manuscript.

Reviewer #2:

Remarks to the Author:

The authors report on the synthesis, structural and electrochemical characterisation of Cu-N-C and Cu-S-N-C materials for oxygen reduction reaction in alkaline electrolyte. The main novel claim is the coordination of Cu by both S and N atoms, not only by N atoms. This seems to improve significantly the ORR activity in alkaline medium of single-atom Copper by tuning the binding of oxygen adsorbates. While demonstrating that the single Cu atoms are coordinated with a mix of N and S atoms versus only N atoms is not an easy task, the authors give convincing evidence for this. The ORR activity in alkaline is at par or higher than state of art Fe-N-C catalysts

Major comments:

a) According to authors, the S L-edge XANES spectrum shows only S-C bonds apparently and no S-Cu signal (suppl. fig 15). Should there not be a specific signal for direct S-Cu binding in this spectrum, if the claim of CuS₁N₃ coordination is correct?

b) The key data that directly supports the Cu-S binding is the EXAFS spectrum at Cu K edge, figure S17. However, it might also be interpreted as superimposed signal from single atom Cu (eg CuN₄) and a small amount of Copper sulfide particles in the sample. Did the authors verify on many locations by STEM that no such particles are present?

c) A puzzling issue is that the calculated XANES signal for CuS₁N₃ is still quite different from the experimental XANES spectrum (Suppl. Figure 20b). The experimental spectrum looks in first approximation maybe more similar to that for crystalline CuS. Could the authors compare the experimental XANES spectrum to that of CuS and other copper sulfide structures in one graph, And try if linear combination fitting of the experimental spectrum with the calculated spectrum for CuN₄ and experimental spectra for CuS and/or Cu₂S gives a possible fit as well, or impossible fit ?

d) On page 6, line 136-138, the authors write that "in-situ environmental microscopic studies suggest that the permeation of sulfur plays an important role for etching the carbon frameworks". It is not clear why and how sulfur would do that. Without addition of sulfur, ZIF-8 (with or without addition of copper salt) leads to high BET area N-C material (e.g. figure S5 shows this, with Cu-ISA/NC having BET of 1377 m²/g, only slightly lower than with sulfur addition). From figure S11, a hole inside the nanozif-8 pyrolyzed particles is seen for S-N-C, but barely seen for Cu/SNC in figure S13

e) The synthesis of S-Cu-ISA/SNC is not fully described in the Methods section on pg 24 and cannot be easily reproduced by others at the moment (eg what amount of sulfur, of Cu-ZIF-8? what size of sulfur particles, etc), what was the ramping rate during heating ?

f) could it be possible to perform Sulfur K-edge xanes and exafs measurements, and compare a sample where Cu-S bonds are expected to exist, to another sample where no such bond is believed to exist (only S-C or S-N) ?

g) last but not least, could the authors comment on the ORR activity in acid medium ? does it also compete with SACs of type Fe-N-C ?

Minor:

Pg 5 line 99: the acronyms used at that location have not yet been explained for Cu-ISA/SNC and Cu-ISA/NC. In particular, how is Cu-ISA/SNC distinct from S-Cu-ISA/SNC? they both contain sulfur...to ease the reading, this should be shortly described there, with reference to full details in Methods section

Pg 6 line 134 : « N₂ absorption-desorption" => adsorption-desorption

Pg 7, line 144 :the atomistic dispersion => atomic dispersion

Figure caption of figure S9 is unclear: "size distribution of single -atom Cu" ...the title says that these are single atoms Cu, while the data is supposed to demonstrate that. Maybe the title could be "The size distribution of Cu signal detected by EDS-HRTEM" ?

Reviewer #3:

Remarks to the Author:

The manuscript, entitled " Engineering unsymmetrical isolated Cu-S1N3 atomic interface for boosting oxygen reduction reaction ", reported that a N, S co-coordinated Cu based single atom electrocatalyst shows an excellent ORR performance. The authors claimed that this kind of asymmetric coordinate structure offers a more suitable free energy and favorable adsorption abilities for intermediates. The work is of significance and novelty in exploring the underlying catalytic mechanism of single metal atomic site such as Cu-S1N3 coordination structure. The experiments are reasonable, and the results are convincing. So I recommend this work to be published in Nature Communications. Before it can be accepted by the journal, there are a few questions needed to be clarified.

1. In Fig 1e, the scale bar should be provided.

2. The ICP data of Zn species should be given to demonstrate its absence in the whole sample. If it exists, a further demonstration of its function in ORR performance should be clarified, as some recent work reported Zn-N₂ active sites may facilitate O-O bond activation (Nature Communications, 2019, DOI: 10.1038/s41467-019-10622-1).

3. The XPS spectrums of S and N species should be provided to analyze the type of these two elements in sample, which can help the rationality of DFT model.

4. Different kinds of S-M- S1N3 ISA/SNC (Mn, Fe, Co, Ni) samples have been synthesized successfully. How about their ORR performance as compared to Cu based N, S co-coordinated structure?

5. The asymmetry of the electronic redistribution discussed in Adv Mater 2019, 1805581 may offer help for discussion in this work.

Responses to the Referees' Comments

We thank the referees for their valuable comments to our manuscript. We have carefully considered the referees' comments and revised the manuscript accordingly. Our responses and corresponding revisions are as follows:

Response to Reviewer 1:

In this work the authors present a CuN_xC_y catalyst containing sulphur atoms and showing an enhanced oxygen reduction reaction activity than the sulphur-free compounds. The morphology and active site structure are mainly investigated by electron microscopy techniques, X-ray absorption spectroscopy and DFT calculations. This is an interesting system, but there are a number of puzzling issues:

Response: Thank you for your positive comments on our manuscript. We have revised our manuscript according to your suggestions.

1) We know that X-ray absorption spectroscopy is a bulk technique, so how the authors can exclude the formation of copper sulphide species coexisting with CuN_x sites, rather than a new compound with formula CuN_xS_y ? This doubt is also fostered by the fact that the Cu-S bond length does not change under working conditions, and if CuS clusters are very small, they cannot be detected by TEM.

Response: Thanks for your insightful question. We agree the viewpoint that X-ray absorption spectroscopy is a bulk technique and if CuS clusters are very small they cannot be detected by conventional TEM. The demonstration of the existence for $\text{Cu-N}_x\text{S}_y$ moieties in S-Cu-ISA/SNC

is a key work in our project, with the applications of macro and micro characterization techniques.

Firstly, it is necessary to mention that the ZIF-8 derived synthetic strategy we applied is a typical and effective method to prepare various metal single atom catalysts, which has been identified by a series of excellent works. The precursor ZIF-8 cages can prevent the aggregation of the encapsulated metal precursor effectively, even at high pyrolytic temperature up to 800-1000 °C.¹⁻⁴

Fig. R1. Representative HAADF-STEM images and the enlarged images of Cu SAs at four different areas in the S-Cu-ISA/SNC sample.

Secondly, as an advanced technique, atomic-resolution high-angle annular dark-field scanning transmission electron microscope (HAADF-STEM) can probe the microstructure and composition of materials at the atomic scale, where contrast is characteristic of atomic number. To elucidate the feature of Cu atoms in S-Cu-ISA/SNC, we carried out aberration corrected HAADF-STEM measurements (**Fig. 1d, e** in the manuscript) with sub-angstrom resolution. Besides, careful examinations at different areas in the sample also confirmed the absence of

small copper sulphide clusters (**Fig. R1**). Then X-ray absorption spectroscopy (**Fig. 2** in the manuscript) provides accurate evidence for the formation of Cu-S and Cu-N bonding on average view. Besides, we also try linear combination fitting (The function is embedded in Athena software) of the experimental XANES spectrum of S-Cu-ISA/SNC with the calculated spectrum for CuN_4 and experimental spectra for CuS and/or Cu_2S , as shown in **Fig. R2**, **Fig. R3** and **Fig. R4**, respectively. The percentage of the three standard components was auto adapted by Athena software. We can find that although the fitted XANES curves near the edge seem coincide with the experimental spectrum of S-Cu-ISA/SNC in some way, the curves after the white line are quite different. In addition, powder X-ray diffraction (PXRD) of the S-Cu-ISA/SNC (**Supplementary Fig. 4b** in the manuscript) reveal only two broad peaks typically for nano-sized graphitic platelets in amorphous carbon materials, suggesting the absence of copper sulphide nanoparticles in the sample. Taking these results into account, it is reliable to demonstrate the formation of CuN_xS_y moieties in S-Cu-ISA/SNC and exclude the existence of copper sulphide species.

Fig. R2. The linear combination fitting of the experimental spectrum of S-Cu-ISA/SNC with the calculated spectrum for CuN_4 and experimental spectrum for CuS.

Fig. R3. The linear combination fitting of the experimental spectrum of S-Cu-ISA/SNC with the calculated spectrum for CuN_4 and experimental spectrum for Cu_2S .

Fig. R4. The linear combination fitting of the experimental spectrum of S-Cu-ISA/SNC with the calculated spectrum for CuN_4 and experimental spectra for CuS and Cu_2S .

Moreover, in order to further exclude the formation of copper sulphide species in S-Cu-ISA/SNC, the sample was immersed in dilute nitric acid (HNO_3) solution (1 mol/L) at 60 °C for 24 h. Due to the fairly high specific surface area and the hierarchically porous characteristics of carbon based frameworks, the dilute nitric acid solution can thoroughly permeate in the whole structure of the S-Cu-ISA/SNC polyhedron, so that the copper sulphide species can be removed if they exist in the sample.⁶ After drying, the acid-treated sample was characterized by XAFS again, which we displayed in **Fig. R5**. It is found that both the XANES (**Fig. R5a**) and EXAFS (**Fig. R5b**) curves are with no obvious change, indicating that the Cu species in S-Cu-ISA/SNC keep the same local atomic structure before and after acid treatment. These results further demonstrate the uniformly isolated Cu species in S-Cu-ISA/SNC and exclude the possible formation of copper sulphide nanoparticles or clusters.

Fig. R5. (a) The XANES spectra and (b) k^3 -weighted FT-EXAFS Cu K-edge spectra of S-Cu-ISA/SNC before and after dilute nitric acid treatment.

Finally, in the manuscript, we describe that “the mean bond length of Cu-S coordination is detected to be nearly unchanged and keeps at about 2.32 Å.” Herein, error bar has to be considered. As we describe in the Supplementary Material, “Error bounds that characterize the structural parameters obtained by EXAFS spectroscopy were estimated as $N \pm 20\%$; $R \pm 1\%$; $\sigma^2 \pm 20\%$; $\Delta E_0 \pm 20\%$ ”. For the bond length which is abstracted from EXAFS, the error bar is ± 0.02 Å. The change of the Cu-S bond length may be slight and it is hard to be monitored by EXAFS (Fig. R6). On the other hand, the nearly unchanged bond length for Cu-S during ORR process just demonstrates the well stability of the Cu-S₁N₃ atomic interface structure at working conditions.

Fig. R6. Bond length evolution of Cu-N and Cu-S for S-Cu-ISA/SNC during ORR process.

References:

1. Chen, Y. et al. Isolated single iron atoms anchored on N-doped porous carbon as an efficient electrocatalyst for the oxygen reduction reaction. *Angew. Chem.* **129**, 7041-7045 (2017).
2. Geng, Z. et al. Achieving a record-high yield rate of $120.9 \mu\text{g}_{\text{NH}_3} \text{mg}_{\text{cat.}}^{-1} \text{h}^{-1}$ for N₂ electrochemical reduction over Ru single-atom catalysts. *Adv. Mater.* **30**, 1803498 (2018).
3. Gu, J. et al. Atomically dispersed Fe³⁺ sites catalyze efficient CO₂ electroreduction to CO. *Science* **364**, 1091-1094 (2019)

4. Xiao, M. et al. Single atom iridium heterogeneous catalyst in oxygen reduction reaction. *Angew. Chem. Int. Ed.* 10.1002/anie.201905241.
5. Xiong, Y. et al. Single-atom Rh/N-doped carbon electrocatalyst for formic acid oxidation. *Nat. Nanotechnol.*, <https://doi.org/10.1038/s41565-020-0665-x> (2020).
6. Jiang, R. et al. Edge-site engineering of atomically dispersed Fe-N₄ by selective C-N bond cleavage for enhanced oxygen reduction reaction activities. *J. Am. Chem. Soc.* **140**, 11594-11598 (2018).

Our revision: We have revised our manuscript according to the reviewer's suggestion (**Line 229-234** in the revised manuscript, **Supplementary Fig. 10, Supplementary Fig. 22 and Supplementary Fig. 27-29**).

2) For the same reason of the previous point, authors cannot claim that a Cu⁺¹ is formed and it is the active center. We know that under applied potential CuNC forms very small Cu nanoparticles (DOI: 10.1038/s41467-018-02819-7; DOI: <https://doi.org/10.1002/anie.201907994>, even if this is the extreme case of very low potential applied). This means that if a small fraction of Cu⁺² is reduced to metallic copper, the average oxidation state will be lower than 2+. It is important to stress that EXAFS is not able to highlight a Cu-Cu contribution if the metallic fraction is small. Still, a peak around 2.5 Å appears in the FT-EXAFS spectrum at 0.75 V, and it is not necessary to find a peak exactly at same position of metallic copper to have a metallic-based cluster with maybe a different nature than that of crystalline copper.

Response: We thank the reviewer for this comment. We read the two papers mentioned by the reviewer carefully (*Nat Commun* **9**, 415 (2018); *Angew. Chem. Int. Ed.* **58**, 15098-15103 (2019)). Both the two previous works are about electroreduction of CO₂ by atomic dispersed Cu catalysts,

and they discovered the same phenomenon that small metallic Cu nanoparticles or clusters formed during CO₂ electrolysis, which was monitored by XAFS measurements at working conditions. Herein, what we have to claim is that oxygen reduction reaction (ORR) is quite a different process from CO₂ reduction reaction (CO₂RR). In our work, CuS₁N₃ species did not form analogous small Cu nanoparticles or clusters under applied potential window during ORR, which had been carefully confirmed by our *ex-situ* XAFS and HAADF-STEM measurements (**Supplementary Fig. 42-43**). Besides, some previous researches also support our viewpoint that the isolated Cu (+1) sites in S-Cu-ISA/SNC may work as the active centers for ORR.¹⁻³ Due to electron transfer between N/S atoms and Cu centers, the aggregation and further reduction of Cu atoms can be prevented effectively.

Moreover, **Fig. R7** shows additional HAADF-STEM images at different areas of the S-Cu-ISA/SNC catalyst after ORR test, which also suggests the absence of Cu clusters or small copper sulphide species. Additionally, our extended *in-situ* XAS measurements we show in **Fig. R12** for **Comment 5** and **Fig. R13** for **Comment 6** also give strong evidence to support our results.

Fig. R7. More HAADF-STEM images of the used S-Cu-ISA/SNC catalyst after ORR test.

In addition, as the reviewer has mentioned, a peak around 2.5 Å appeared in the FT-EXAFS spectrum at 0.75 V vs RHE. In our opinion, it is not the signal originated from Cu-Cu. When we changed the FT range from “2.0-11.0 Å⁻¹” to “2.0-10.0 Å⁻¹”, the peak around 2.5 Å was erased, while the main peak around 1.5 Å almost kept the same (**Fig. R8**). If the peak at 2.5 Å is attributed to Cu-Cu back scattering, there is no reason that it is such sensitive to the slight change of FT range. The weak signal at the range 2.2 to 4.2 Å may be derived from Cu-C/N back scattering at the high atomic shells and may also mix with some noise signals. It is hard for us to abstract useful structure information from this range.

Fig. R8. Cu K-edge FT-EXAFS spectra of S-Cu-ISA/SNC at 0.75 V vs RHE in different FT windows (0.75 V-1: 2.0-11.0 Å⁻¹, 0.75 V-2: 2.0-10.0 Å⁻¹).

References:

1. Wu, H. et al. Highly doped and exposed Cu(i)-N active sites within graphene towards efficient oxygen reduction for zinc-air batteries. *Energy Environ. Sci.* **9**, 3736-3745 (2016).
2. Iwase, K. et al. Copper-modified covalent triazine frameworks as non-noble-metal electrocatalysts for oxygen reduction. *Angew. Chem. Int. Ed.* **54**, 11068-11072 (2015).

3. Gentil, S. et al. Electrocatalytic O₂ reduction at a bio-inspired mononuclear copper phenolato complex immobilized on a carbon nanotube Electrode. *Angew. Chem. Int. Ed.* **55**, 2517-2520 (2016).

Our revision: We have revised our manuscript according to the reviewer's suggestion (**Line 359-361** in the revised manuscript, **Supplementary Fig. 59**).

3) Therefore, on the basis of the issue 1) and 2) the DFT model proposed by the authors may not be valid.

Response: Based on the response for issue 1 and issue 2, we demonstrate the uniformly isolated Cu species in S-Cu-ISA/SNC and exclude the formation of copper sulphide particles or clusters through STEM and XAFS analysis. Besides, our XPS and soft XAS measurements (**Fig. 2** and **Supplementary Fig. 16**.) also provide evidences for the formation of Cu-N_xS_y species in S-Cu-ISA/SNC. According the above results, we propose the unsymmetrical isolated Cu-S₁N₃ atomic interface structure for S-Cu-ISA/SNC, also considering other possible structure model as we have illustrated in the **Fig. R9 (Supplementary Fig. 60)**. So it is reliable that the proposed DFT model is valid.

Fig. R9. Atomic structures schematics of two kinds of S-doping models and two kinds of Cu-S₂N₂ considered in formation energy calculation. (a) Cu-S₁N₃, in which a doped S atom substitutes only one atom of C or N, and (b) Cu-S₁N₃, in which a doped S atom substitutes two adjacent atoms of C and N. (c) Cu-para-S₂N₂, (d) Cu-ortho-S₂N₂. Gray, blue, orange and yellow balls represent C, N, Cu and S atoms, respectively.

Other minor issues:

4) Why for the estimation of the oxidation number of the catalyst, only Cu foil and CuO references are used (figure 16 of supplementary materials)? This kind of determination requires attention, because the oxidation state (related to the XAS edge position) is not absolute, and it depends on the specific compound.

Response: We agree the reviewer's viewpoint that the oxidation state (related to the XAS edge position) is not absolute, and it depends on the specific compound. As we known, metal foil and metal oxides (such as Cu foil and CuO) are commonly used standards for XANES and EXAFS analysis in various research works, mainly due to their fine stability and easy achievement.¹⁻⁴

Based on the viewpoint that the edge position in the XANES spectra reflects the oxidation state of Cu in different samples,⁵ we adopt Cu foil, Cu₂O, CuO, Cu₂S, CuS and CuPc as references for the analysis of *ex-situ* XANES spectra more comprehensively, as shown in **Fig. R10**. The fitted oxidation state of Cu in fresh S-Cu-ISA/SNC sample from XANES spectra is 1.97. Moreover, the oxidation state evolution of Cu species at working conditions from the *in-situ* XANES curves is also displayed in **Fig. R11**.

Fig. R10. (a-b) The Cu K-edge XANES spectra of S-Cu-ISA/SNC and the references (Cu foil, Cu₂O, CuO, Cu₂S, CuS and CuPc). (c) First-derivative XANES curves of S-Cu-ISA/SNC and the references. (d) Correlation between the Cu oxidation state and the energy position of the XANES spectrum, determined as the first maximum of the first derivative spectrum of S-Cu-ISA/SNC and different copper reference compounds.

Fig. R11. (a) The fitted average oxidation states of Cu from XANES spectra. It was observed that the average valence state of Cu decreased from +1.94 to +1.14 under working conditions (from 1.05 V to 0.75 V vs RHE). (b) Current density as a function of potential for S-Cu-ISA/SNC (left) and the average oxidation number of Cu species in S-Cu-ISA/SNC as a function of potential (right).

References:

1. Yang, H., et al. Scalable production of efficient single-atom copper decorated carbon membranes for CO₂ electroreduction to methanol, *J. Am. Chem. Soc.* **141**, 12717-12723 (2019).
2. Zheng, X., Ji, Y., Tang, J. et al. Theory-guided Sn/Cu alloying for efficient CO₂ electroreduction at low overpotentials. *Nat Catal.* **2**, 55-61 (2019).
3. Weng, Z., Wu, Y., Wang, M. et al. Active sites of copper-complex catalytic materials for electrochemical carbon dioxide reduction. *Nat Commun.* **9**, 415 (2018).
4. Fang, S., et al. Uncovering near-free platinum single-atom dynamics during electrochemical hydrogen evolution reaction. *Nat Commun.* **11**, 1029 (2020).

5. Karapinar, D. et al. Electroreduction of CO₂ on single-site copper-nitrogen-doped carbon material: selective formation of ethanol and reversible restructuring of the metal sites, *Angew. Chem. Int. Ed.* **58**, 15098-15103 (2018).

Our revision: We have revised our manuscript according to the reviewer's suggestion (**Fig. 4d** in the revised manuscript, **Supplementary Fig. 20** and **Supplementary Fig. 51**).

5) Why XAS spectra are just recorded between 0.75 and 1.05 V?

Response: Thanks for this important question. Considering that the onset potential (E_{onset}) of S-Cu-ISA/SNC is 1.05 V and that 0.75 V just situates around the limited current stage for ORR in 0.1 M KOH, the XAS spectra recorded between 0.75 and 1.05 V can typically reflect the variation of catalytic active sites under working conditions.¹⁻³

Fig. R12. Cu K-edge XANES spectra of S-Cu-ISA/SNC at 0.6 V and 0.4 V compared to that at 0.75 V vs RHE.

In addition, we carried out extended *in-situ* XAS measurements at lower potentials at BL14W1 in the Shanghai Synchrotron Radiation Facility (SSRF). The XANES spectra at 0.6 V

and 0.4 V vs RHE were recorded (**Fig. R12**). We can find that the spectra at 0.6 V and 0.4 V vs RHE show little change compared to that at 0.75 V (particularly the absorption edge and white line peak as shown in **Fig. R12b** and **Fig. R12c**, respectively), which indicates that the local atomic structure and oxidation state of Cu in S-Cu-ISA/SNC at 0.6 V and 0.4 V during ORR is just the same as that at 0.75 V, mainly due to electron transfer between N/S atoms and Cu centers. This also means that the average oxidation state of Cu species ($\text{Cu-S}_1\text{N}_3$) keeps stable (around +1) when the potential is down to 0.75 V, rather than further reduce tends to zero (metallic copper). The extended *in-situ* XAFS measurements give strong evidences that no Cu particles or clusters form at the applied potentials and also demonstrate that that Cu (+1) sites in S-Cu-ISA/SNC may work as the active centers for ORR.

References:

1. Zitolo, A., Ranjbar-Sahraie, N., Mineva, T. et al. Identification of catalytic sites in cobalt-nitrogen-carbon materials for the oxygen reduction reaction. *Nat Commun.* **8**, 957 (2017).
2. Chen Y, Ji S, Zhao S, et al. Enhanced oxygen reduction with single-atomic-site iron catalysts for a zinc-air battery and hydrogen-air fuel cell. *Nat. Commun.* **9**, 5422 (2108).
3. Li J, Ghoshal S, Liang W, et al. Structural and mechanistic basis for the high activity of Fe-N-C catalysts toward oxygen reduction. *Energy Environ. Sci.* **9**, 2418-2432 (2016).

Our revision: We have revised our manuscript according to the reviewer's suggestion (**Line 333** in the revised manuscript, **Supplementary Fig. 52**).

6) Is the *in-situ* XAS spectrum reversible when you come back to the OCV potential?

Response: We appreciate the reviewer for the insightful point, which has helped us much to improve the quality of our manuscript. We performed additional *in-situ* XAS measurements to

evaluate the reversibility of Cu K-edge XANES spectra of S-Cu-ISA/SNC at various potentials (Fig. R13). As what we have described in Fig. 4b and Supplementary Fig. 50, from 1.05 V to 0.75 V, the absorption edge of Cu is gradually shifted to the lower energy, along with reduction of the white line peak, meaning a decrease of the average oxidation state of Cu in S-Cu-ISA/SNC during ORR. Interestingly, when the applied potential returned to OCV (from 0.75 V to 1.05 V), Cu XANES edge shifted back to higher energy along with increase of the white line peak (Fig. R13). This provides unequivocal evidence that the XAS spectra as a function of applied potential are reversible, which might be due to the strong anchor effect of N and S atom to the Cu sites. The reversible change of Cu valence state indicated the active contribution of isolated Cu atoms in the catalytic reaction for ORR.

Fig. R13. (a) Cu K-edge XANES spectra of S-Cu-ISA/SNC at various potentials during ORR catalysis in O₂-saturated 0.1 M KOH from 0.75 V back to 1.05 V. (b) the absorption edge evolution and (c) the white line peak evolution of S-Cu-ISA/SNC at Cu K-edge under *in-situ* XAFS condition; (d) Differential $\Delta\mu$ XANES spectra obtained by subtracting the normalized spectrum at every potential to the spectrum recorded at 0.75 V vs RHE.

Fig. R14. The fitted average oxidation states of Cu from XANES spectra from 0.75 V to 1.05 V vs RHE. It was observed that the average valence state of Cu increase from +1.12 to +1.90 under working conditions.

Our revision: We have revised our manuscript according to the reviewer's suggestion (**Line 335-340** in the revised manuscript, **Supplementary Fig. 53-54**).

7) I personally do not appreciate images like Figure 1e with red circles around hypothetical sites. In many parts of these pictures the spots overlap and the hypothetical single-atoms site is not clear.

Response: Thank you for your kind suggestions. We have erased the red circles around the bright dots in **Fig R15**. Moreover, we also modified other STEM images with red circles in the manuscript and supplementary materials.

Fig. R15. Morphology and composition characterizations of S-Cu-ISA/SNC. (a) SEM, (b) TEM and (c) EDS images of S-Cu-ISA/SNC, C (pink), N (green), S (yellow) and Cu (red). (d) HAADF-STEM image and (e) the magnified image of S-Cu-ISA/SNC. (f) The corresponding intensity profiles along the line X-Y in e.

Our revision: We have modified the manuscript according to the reviewer's suggestions (**Fig. 1e**, **Fig. 6a**, **Fig. 6c**, **Fig. 6e** and **Fig. 6g** in the revised manuscript, **Supplementary Fig. 14d**, **Supplementary Fig. 15d** and **Supplementary Fig. 42d**).

8) The authors completely forget to mention a lot of literature from other worldwide research groups, and not only from the restricted geographical area of most of the authors in the bibliography of this manuscript.

Response: Thanks to the reviewer's valuable suggestion. We have added some suitable references in the revised manuscript.

The added references:

1. Fan, L. et al. Atomically isolated nickel species anchored on graphitized carbon for efficient hydrogen evolution electrocatalysis. *Nat. Commun.* **7**, 10667 (2016).
2. Zhang, L. et al. Graphene defects trap atomic Ni species for hydrogen and oxygen evolution reactions. *Chem* **4**, 285–297 (2018).
3. Weng, Z. et al. Electrochemical CO₂ reduction to hydrocarbons on a heterogeneous molecular Cu catalyst in aqueous solution. *J. Am. Chem. Soc.* **138**, 8076-8079 (2016).
4. Copéret, C. et al. Surface organometallic and coordination chemistry toward single-site heterogeneous catalysts: strategies, methods, structures, and activities. *Chem. Rev.* **116**, 323-421 (2016).
5. Elgrishi, N. et al. Molecular polypyridine-based metal complexes as catalysts for the reduction of CO₂. *Chem. Soc. Rev.* **46**, 761-796 (2017).
6. Yuan, K. et al. Boosting oxygen reduction of single iron active sites via geometric and electronic engineering: nitrogen and phosphorus dual-coordination. *J. Am. Chem. Soc.* **142**, 2404-2412 (2020).
7. Lee, S. H. et al. Design principle of Fe-N-C electrocatalysts: how to optimize multimodal porous structures? *J. Am. Chem. Soc.* **141**, 2035-2045 (2019).

8. Wagner, S. et al. Elucidating the structural composition of a Fe-N-C catalyst by nuclear- and electron-resonance techniques. *Angew. Chem. Int. Ed.* **58**, 10486-10492 (2019).
9. Cheng, C. et al. Atomic Fe-N_x coupled open-mesoporous carbon nanofibers for efficient and bioadaptable oxygen electrode in Mg-air batteries. *Adv. Mater.* **30**, 1802669 (2018).
10. Zitolo, A. et al. Identification of catalytic sites in cobalt-nitrogen-carbon materials for the oxygen reduction reaction. *Nat. Commun.* **8**, 957 (2017).
11. Sa, Y. J. et al. A general approach to preferential formation of active Fe-N_x sites in Fe-N/C electrocatalysts for efficient oxygen reduction reaction. *J. Am. Chem. Soc.* **138**, 15046-15056 (2016).
12. Zhuang, L., et al. Defect-induced Pt-Co-Se coordinated sites with highly asymmetrical electronic distribution for boosting oxygen-involving electrocatalysis. *Adv. Mater.* **31**, 1805581 (2019).
13. Bakandritsos, A. et al. Mixed-valence single-atom catalyst derived from functionalized graphene. *Adv. Mater.* **31**, 1900323 (2019).
14. Sun, Y. et al. Activity-selectivity trends in the electrochemical production of hydrogen peroxide over single-site metal-nitrogen-carbon catalysts. *J. Am. Chem. Soc.* **141**, 12372-12381 (2019).
15. Li, F. et al. Identifying the structure of Zn-N₂ active sites and structural activation. *Nat. Commun.* **10**, 2623 (2019).
16. Weng, Z., et al. Active sites of copper-complex catalytic materials for electrochemical carbon dioxide reduction. *Nat Commun* **9**, 415 (2018).

17. Karapinar, D. et al. Electroreduction of CO₂ on single-site copper-nitrogen-doped carbon material: selective formation of ethanol and reversible restructuring of the metal sites, *Angew. Chem. Int. Ed.* **58**, 15098-15103 (2019).

Our revision: We have revised our manuscript according to the reviewer's suggestion (**References 14-18, Reference 24-29, Reference 44-46, Reference 52, and References 65-66** in the revised manuscript).

Response to Reviewer 2 :

The authors report on the synthesis, structural and electrochemical characterisation of Cu-N-C and Cu-S-N-C materials for oxygen reduction reaction in alkaline electrolyte. The main novel claim is the coordination of Cu by both S and N atoms, not only by N atoms. This seems to improve significantly the ORR activity in alkaline medium of single-atom Copper by tuning the binding of oxygen adsorbates. While demonstrating that the single Cu atoms are coordinated with a mix of N and S atoms versus only N atoms is not an easy task, the authors give convincing evidence for this. The ORR activity in alkaline is at par or higher than state of art Fe-N-C catalysts.

Response: Thank you for your time in reviewing this manuscript. We have carefully considered your comments and revised our manuscript according to your suggestions.

Major comments:

a) According to authors, the S L-edge XANES spectrum shows only S-C bonds apparently and no S-Cu signal (suppl. fig 15). Should there not be a specific signal for direct S-Cu binding in this spectrum, if the claim of CuS_1N_3 coordination is correct?

Response: Thanks for your insightful question. In the manuscript, we claim that “The S L-edge XANES spectrum of S-Cu-ISA/SNC shows obvious peaks (peak h-j)) in the region of 163~167 eV corresponding to C-S-C coordination species, suggesting the anchor of S in the carbon skeleton.” Herein, we did not deny the existence of S-Cu signal. On the contrary, there are some other peaks in the 165~185 eV region, which are probably attributable to the S-Cu bonds (**Fig.**

R16.)¹⁻³ The local structure of S species can be reflected more clearly by S K-edge EXAFS spectrum as shown in **Fig. R25** for **Comment f**.

Fig. R16. S L-edge XANES spectrum of the S-Cu-ISA/SNC.

References:

1. Kasrai, M. et al. A XANES study of the S L-edge in sulfide minerals: Application to interatomic distance determination. *Solid State Communications*, **68**, 507-511 (1988).
2. Li, D. et al. S K- and L-edge XANES and electronic structure of some copper sulfide minerals. *Phys Chem Minerals* **21**, 317-324 (1994).
3. Kasrai, M. et al. Sulphur characterization in coal from X-ray absorption near edge spectroscopy. *International Journal of Coal Geology* **32**, 107-135 (1996).

Our revision: We have revised our manuscript according to the reviewer's suggestion (**Supplementary Fig. 17**).

b) The key data that directly supports the Cu-S binding is the EXAFS spectrum at Cu K edge, figure S17. However, it might also be interpreted as superimposed signal from single atom Cu

(eg CuN_4) and a small amount of Copper sulfide particles in the sample. Did the authors verify on many locations by STEM that no such particles are present?

Response: Thanks for your insightful question. Atomic-resolution high-angle annular dark-field scanning transmission electron microscope (HAADF-STEM) can properly probe the microstructure and composition of materials at the atomic scale. To further elucidate the formation of the Cu atoms, we carried out aberration corrected HAADF-STEM measurements at four different areas for the S-Cu-ISA/SNC samples. The STEM images confirmed the absence of small copper sulphide particles or clusters (**Fig. R17**). By comprehensive consideration of the STEM and EXAFS results, it is reliable to claim that the Cu-S bonds originate from $\text{Cu-N}_x\text{S}_y$ moieties anchored in the carbon matrix but not from copper sulfide particles in the sample.

Fig. R17. Representative HAADF-STEM images of Cu SAs at four different areas for the S-Cu-ISA/SNC sample.

In addition, it is necessary to mention that the ZIF-8 derived synthetic strategy we applied is a typical and effective method to prepare various metal single atom catalysts, which has been

identified by a series of excellent works. The precursor ZIF-8 cages can prevent the aggregation of the encapsulated metal ions efficiently, even at high pyrolytic temperature of 800-1000 °C.¹⁻⁴

In order to further exclude the formation of copper sulphide species in S-Cu-ISA/SNC, the sample was immersed in dilute nitric acid (HNO₃) solution (1 mol/L) at 60 °C for 24 h. Due to the fairly high specific surface area and the hierarchically porous characteristics of carbon based frameworks, the dilute nitric acid solution can thoroughly permeate in the whole structure of the S-Cu-ISA/SNC polyhedron, so that the copper sulphide species can be removed if they exist in the sample.⁶ After drying, the acid-treated sample was characterized by XAFS again, which we displayed in **Fig. R18**. It is found that both the XANES (**Fig. R18a**) and EXAFS (**Fig. R18b**) curves are with no obvious change, indicating that the Cu species in S-Cu-ISA/SNC keep the same local atomic structure before and after acid treatment. These results further demonstrate the uniformly isolated Cu species in S-Cu-ISA/SNC and exclude the possible formation of copper sulphide nanoparticles or clusters.

Fig. R18. (a) The XANES spectra and (b) k^3 -weighted FT-EXAFS Cu K-edge spectra of S-Cu-ISA/SNC before and after dilute nitric acid treatment.

References:

1. Chen, Y. et al. Isolated single iron atoms anchored on N-doped porous carbon as an efficient electrocatalyst for the oxygen reduction reaction. *Angew. Chem.* **129**, 7041-7045 (2017).
2. Geng, Z. et al. Achieving a record-high yield rate of $120.9 \mu\text{g}_{\text{NH}_3} \text{mg}_{\text{cat.}}^{-1} \text{h}^{-1}$ for N_2 electrochemical reduction over Ru single-atom catalysts. *Adv. Mater.* **30**, 1803498 (2018).
3. Gu, J. et al. Atomically dispersed Fe^{3+} sites catalyze efficient CO_2 electroreduction to CO. *Science* **364**, 1091-1094 (2019)
4. Xiao, M. et al. Single atom iridium heterogeneous catalyst in oxygen reduction reaction. *Angew. Chem. Int. Ed.* **58**, 9640-9645 (2019).
5. Xiong, Y. et al. Single-atom Rh/N-doped carbon electrocatalyst for formic acid oxidation. *Nat. Nanotechnol.* <https://doi.org/10.1038/s41565-020-0665-x> (2020).
6. Jiang, R. et al. Edge-site engineering of atomically dispersed Fe-N₄ by selective C-N bond cleavage for enhanced oxygen reduction reaction activities. *J. Am. Chem. Soc.* **140**, 11594-11598 (2018).

Our revision: We have revised our manuscript according to the reviewer's suggestion (**Supplementary Fig. 10** and **Supplementary Fig. 22**).

c) A puzzling issue is that the calculated XANES signal for CuS_1N_3 is still quite different from the experimental XANES spectrum (Suppl. Figure 20b). The experimental spectrum looks in first approximation maybe more similar to that for crystalline CuS. Could the authors compare the experimental XANES spectrum to that of CuS and other copper sulfide structures in one graph, and try if linear combination fitting of the experimental spectrum with the calculated spectrum for CuN_4 and experimental spectra for CuS and/or Cu_2S gives a possible fit as well, or impossible fit?

Response: Thanks for the reviewer's insightful suggestion. As we known, for XANES calculations, due to the variable electronic potential and broadening, it is really hard or even impossible to obtain a simulated XANES spectrum which is 100% identical to the experimental one with no difference.¹⁻³ In the **Supplementary Fig. 25b** in the revised supplementary materials, the calculated result based on Cu-S₁N₃ model can perfectly reproduce the five features (A, B, C, D and E) of the experimental curve for S-Cu-ISA/SNC, strongly indicating that Cu-S₁N₃ is the optimum configuration.

Furthermore, we compare the experimental XANES spectrum of S-Cu-ISA/SNC to that of CuS and Cu₂S in one graph (**Fig. R19**). We can see that the XANES curve of S-Cu-ISA/SNC is quite different from those of CuS and Cu₂S, which further provide evidences for the absence of copper sulphide species in the sample.

Fig. R19. The Cu K-edge XANES spectra of S-Cu-ISA/SNC and the references (Cu₂S and CuS).

Moreover, we also try linear combination fitting (The function is embedded in Athena software) of the experimental XANES spectrum of S-Cu-ISA/SNC with the calculated spectrum for CuN₄ and experimental spectra for CuS and/or Cu₂S, as shown in **Fig. R20**, **Fig. R21** and **Fig.**

R22, respectively. The percentage of the three standard components was auto adapted by Athena software. We can find that although the fitted XANES curves near the edge seem coincide with the experimental spectrum of S-Cu-ISA/SNC in some way, the curves after the white line are quite different. Based on the above analysis, we can exclude the possible existence of copper sulfide species in S-Cu-ISA/SNC.

Fig. R20. The linear combination fitting of the experimental spectrum of S-Cu-ISA/SNC with the calculated spectrum for CuN_4 and experimental spectrum for CuS .

Fig. R21. The linear combination fitting of the experimental spectrum of S-Cu-ISA/SNC with the calculated spectrum for CuN₄ and experimental spectrum for Cu₂S.

Fig. R22. The linear combination fitting of the experimental spectrum of S-Cu-ISA/SNC with the calculated spectrum for CuN₄ and experimental spectra for CuS and Cu₂S.

References:

1. Zitolo, A. et al. Identification of catalytic sites for oxygen reduction in iron- and nitrogen-doped graphene materials. *Nat. Mater.* **14**, 937-942 (2015).
2. Liu, W. et al. Single-atom dispersed Co-N-C catalyst: structure identification and performance for hydrogenative coupling of nitroarenes. *Chem. Sci.* **7**, 5758-5764 (2016).
3. Fei, H. et al. General synthesis and definitive structural identification of MN₄C₄ single-atom catalysts with tunable electrocatalytic activities. *Nat. Catal.* **1**, 63-72 (2018).
4. Xiong, Y. et al. Single-atom Rh/N-doped carbon electrocatalyst for formic acid oxidation. *Nat. Nanotechnol.* <https://doi.org/10.1038/s41565-020-0665-x> (2020).

Our revision: We have revised our manuscript according to the reviewer's suggestion (**Line 229-234** in the revised manuscript, **Supplementary Fig. 27-29**).

d) On page 6, line 136-138, the authors write that “in-situ environmental microscopic studies suggest that the permeation of sulfur plays an important role for etching the carbon frameworks”. It is not clear why and how sulfur would do that. Without addition of sulfur, ZIF-8 (with or without addition of copper salt) leads to high BET area N-C material (e.g. figure S5 shows this, with Cu-ISA/NC having BET of 1377 m²/g, only slightly lower than with sulfur addition). From figure S11, a hole inside the nano zif-8 pyrolyzed particles is seen for S-N-C, but barely seen for Cu/SNC in figure S13.

Response: Thanks for insightful comments. To investigate the role of sulfur, the *in-situ* environmental microscopic studies were carried out to simulate the variation of carbon frameworks with temperature. From **Supplementary Fig. 7** and **Supplementary Video 1**, it can be seen that carbon frameworks became shrunken following the increasing of temperature. Especially, the smaller particle disappeared when the temperature was up to 980°C. Interestingly, carbon frameworks without sulfur retained their shape, but just became smaller with the increasing temperature mainly due to the evaporation of Zn (**Supplementary Fig. 8**, **Supplementary Video 2**). The results confirm that sulfur can etch the carbon frameworks effectively. Moreover, the entire carbon frameworks can be completely eroded if the sulfur amount is enough. In this work, we optimize the amount of sulfur to ensure that the carbon frameworks are moderately etched to form more porous space. The adsorption-desorption curve was further performed to evaluate the effect of sulfur to the carbon frameworks (**Supplementary Fig. 6**). The results show that S-Cu-ISA/SNC (with addition of sulfur) possesses ~20% higher

BET area than that of Cu-ISA/NC (without addition of sulfur). Particularly, S-Cu-ISA/SNC obtain much larger total pore volume between 0.5 nm to 30 nm than that of Cu-ISA/NC ($2.13 \text{ cm}^3 \text{ g}^{-1}$: $0.67 \text{ cm}^3 \text{ g}^{-1}$), indicating the prominent etching effect of sulfur to construct the hierarchically porous structure for the carbon frameworks. In addition, **Fig. R23** shows additional TEM images for the Cu-ISA/SNC. A hole inside the carbon matrix is obviously observed for Cu-ISA/SNC, just similar to the SNC sample in revised **Supplementary Fig. 13**.

Fig. R23. The additional TEM images of the Cu-ISA/SNC sample.

Our revision: We have revised our manuscript according to the reviewer's suggestion (**Supplementary Fig. 15a**).

e) The synthesis of S-Cu-ISA/SNC is not fully described in the Methods section on pg 24 and cannot be easily reproduced by others at the moment (eg what amount of sulfur, of Cu-ZIF-8? what size of sulfur particles, etc), what was the ramping rate during heating?

Response: Thanks for this comment. Actually, the mass ration of sulfur powder and Cu-ZIF-8 is 1: 10. The sulfur powder (99.5%, Alfa) is 325 mesh (sulfur powder can be dissolved in carbon tetrachloride mixed solution to ensure sulfur species adsorb on MOF particles) and the ramping

rate during heating is 5 °C/min. Moreover, the SEM images of the S-Cu-ZIF-8 sample were exhibited in **Fig. R24**. It is observed that the sulfur was absorbed on the surface of Cu-ZIF-8 powder with small size (even below 10 nm). Besides, some large sulfur particles with size of several hundred nanometers are also detected.

Fig. R24. SEM images of the S-Cu-ZIF-8 sample.

Our revision: We have provided more detailed synthesis procedure of S-Cu-ISA/SNC in the revised manuscript and supplementary materials (**Line 486**, **Line 495** and **Line 498-499** in the revised manuscript, **Supplementary Fig. 2**).

f) Could it be possible to perform Sulfur K-edge xanes and exafs measurements, and compare a sample where Cu-S bonds are expected to exist, to another sample where no such bond is believed to exist (only S-C or S-N)?

Response: Thanks for the reviewer's insightful question. We carried out XANES and EXAFS measurements at S K-edge to further investigate the local atomic structure of S-Cu-ISA/SNC. The XANES and EXAFS spectra of S K-edge were recorded at the 4B7A station in Beijing

Synchrotron Radiation Facility in TEY mode. The samples were coated on double-sided carbon tape for characterization.

The valance state of sulfur was investigated by S K-edge XANES, applied CuS (-2), S powder (0), Na₂SO₃ (+4) and Na₂SO₄ (+6) as references (**Fig. R25**). Generally, the oxidation states of S are linear correlated to the position of K-edge.¹⁻³ We found that the sulfur in S-Cu-ISA/SNC appeared as slightly positive charge. These results indicated the origin of positive valence S atoms in S-Cu-ISA/SNC might be attributed to the existence of their coordination with N atoms, since N has stronger electronegativity (3.04) compared to S (2.58), as well as the existence of C-SO_x species in the sample.

Furthermore, in **Fig. R26**, the S K-edge EXAFS for S-Cu-ISA/SNC demonstrated the presence of S-Cu bonding, compared with CuS reference (Cu-S bonds are expected to exist) and SNC sample (no Cu-S bond is believed to exist, only S-C or S-N).

Fig. R25. S K-edge XANES spectra of S-Cu-ISA/SNC and reference materials.

Fig. R26. Fourier transformed magnitudes of the experimental S K-edge EXAFS signals of S-Cu-ISA/SNC, CuS and SNC. It can be observed that, in comparison with CuS and SNC, S-Cu-ISA/SNC shows an additional peak at 2.1 Å, which can be ascribed to the presence of S-Cu nearest-neighboring coordination.

References:

1. Afanasiev, P., et al. Low-temperature hydrogen interaction with amorphous molybdenum sulfides MoS_x . *J. Phys. Chem. C* **113**, 4139-4146 (2009).
2. Ha, Y., et al. Sulfur K-edge X-ray absorption spectroscopy and density functional theory calculations on monooxo Mo~IV and bisoxo Mo~VI bisdithiolenes: insights into the mechanism of oxo transfer in sulfite oxidase and its relation to the mechanism of DMSO reductase. *J. Am. Chem. Soc.* **136**, 9094-9105 (2014).
3. Henthorn, J., et al. Localized electronic structure of nitrogenase FeMoco re-vealed by selenium K-edge high resolution X-ray absorption spectroscopy. *J Am Chem Soc.* **141**, 13676-13688 (2019).

Our revision: We have revised our manuscript according to the reviewer's suggestion (**Line 184-191** in the revised manuscript, **Supplementary Fig. 18** and **Supplementary Fig. 19**).

g) last but not least, could the authors comment on the ORR activity in acid medium? Does it also compete with SACs of type Fe-N-C?

Response: We thank the reviewer for this comment. We have evaluated the ORR performance of S-Cu-ISA/SNC and reference catalysts in the acidic media by measuring in 0.5 M H₂SO₄ solution. **Fig. R27a** show the ORR polarization curves of different catalysts. S-Cu-ISA/SNC exhibited the highest ORR activity among the three Cu-based catalysts (S-Cu-ISA/SNC, Cu-ISA/SNC and Cu-ISA/NC), with onset (E_{onset}) and half-wave ($E_{1/2}$) potentials of 0.86 V and 0.74 V, respectively. The $E_{1/2}$ of S-Cu-ISA/SNC was just 80 mV lower than that of commercial Pt/C (0.82 V), which is widely used as a benchmark. As we can see, the ORR catalytic activity of S-Cu-ISA/SNC was comparable to Fe-N-C catalysts reported in the literatures under acid conditions (**Table R1**). The Tafel plots exhibited in **Fig. R27b** confirmed the favorable ORR kinetics of S-Cu-ISA/SNC with low Tafel slope of 106.9 mV dec⁻¹. Koutecky-Levich (K-L) plots of S-Cu-ISA/SNC were derived from linear sweep voltammetry (LSV) curves (**Fig. R27c**). The electron transfer number (n) of S-Cu-ISA/SNC was 3.58-3.73 (**Fig. R27d**), indicating a direct four-electron ORR pathway in acidic conditions. As shown in the **Fig. R27e**, the electron transfer number of S-Cu-ISA/SNC was about 3.90 and the H₂O₂ yield remained below 7%, suggesting its high-efficiency 4e⁻ ORR pathway. Moreover, the accelerated durability of S-Cu-ISA/SNC revealed the outstanding stability with little negative in $E_{1/2}$ after 5000 cycles (**Fig. R27f**).

Fig. R27. Electrocatalytic ORR performance of S-Cu-ISA/SNC in 0.5 M H₂SO₄ solution and 20% Pt/C in 0.1 M HClO₄ solution. (a) ORR polarization curves for S-Cu-ISA/SNC, Cu-ISA/SNC, Cu-ISA/NC, SNC, NC and 20% Pt/C. (b) The Tafel plots for S-Cu-ISA/SNC and the corresponding reference catalysts from the data in panel a. (c) The ORR polarization curves of S-Cu-ISA/SNC at different rotating rates. (d) The K-L plots and electron-transfer numbers for S-Cu-ISA/SNC. (e) Electron transfer number (*n*) (top) and H₂O₂ yield (bottom) vs potential. (f) ORR polarization curves of S-Cu-ISA/SNC before and after 5000 potential cycles.

Table R1. Comparison of ORR performance between S-Cu-ISA/SNC and SACs of type Fe-N-C catalysts reported in the literatures under acid electrolyte.

Electrocatalysts	Electrolyte	Onset potential (V vs. RHE)	Half-wave potential (V vs. RHE)	Limited current (mg cm ⁻²)	Reference
S-Cu-ISA/SNC	0.5M H ₂ SO ₄	0.86	0.74	4.06	This work
Fe-SAs/NPS-HC	0.5M H ₂ SO ₄	0.91	0.791	5.01	Nat. Commun. 9 , 5422 (2018).
Fe/SNC	0.5M H ₂ SO ₄	0.89	0.77	4.80	Angew. Chem. Int. Ed. 56 , 13800-13804 (2017).
SA-Fe/NG	0.5M H ₂ SO ₄	0.90	0.80	5	PNAS. 115 , 6626-6631 (2018).
Fe-N-CNF	0.5M H ₂ SO ₄	0.84	0.62	5.2	Angew. Chem. Int. Ed. 54 , 8179 (2015).
Fe-ISAs/CN	0.1M HClO ₄	0.9	0.79	5.8	Angew. Chem. Int. Ed. 56 , 6937-6941 (2017).
Fe-N/C-800	0.1M HClO ₄	0.82	0.6	6.09	J. Am. Chem. Soc. , 136 , 11027-11033 (2014).
Fe-CNT/PC	0.1M HClO ₄	0.95	0.79	5.9	J. Am. Chem. Soc. 2016 , 138, 15046-15056.
CPANI-Fe-NaCl	0.1M HClO ₄	0.88	0.73	5	J. Am. Chem. Soc. 137 , 5414-5420 (2015).

Our revision: We have added the ORR activity of S-Cu-ISA/SNC in acid medium in the revised manuscript. (Line 289-294 in the revised manuscript, **Supplementary Fig. 44** and **Supplementary Table 3**)

Minor:

Pg 5 line 99: the acronyms used at that location have not yet been explained for Cu-ISA/SNC and Cu-ISA/NC. In particular, how is Cu-ISA/SNC distinct from S-Cu-ISA/SNC? They both

contain sulfur...to ease the reading, this should be shortly described there, with reference to full details in Methods section.

Response: We thank the reviewer for this comment. We explain the acronyms used for Cu-ISA/NC, Cu-ISA/SNC and S-Cu-ISA/SNC as follow:

Cu-ISA/NC: isolated single-atom Cu-N₄ supported on N doped carbon polyhedron;

Cu-ISA/SNC: isolated single-atom Cu-N₄ supported on N and S co-doped carbon polyhedron;

S-Cu-ISA/SNC: isolated single-atom Cu-S₁N₃ supported on N and S co-doped carbon polyhedron.

The sentence has been added in the revised manuscript as follow: “The achieved S-Cu-ISA/SNC plays a boosted ORR performance in alkaline media with a half-wave potential 0.918 V vs RHE, compared to the counterparts, following the activity trend S-Cu-ISA/SNC (**isolated single-atom Cu-S₁N₃ supported on N and S co-doped carbon polyhedron**) > Cu-ISA/SNC (**isolated single-atom Cu-N₄ supported on N and S co-doped carbon polyhedron**) > Cu-ISA/NC (**isolated single-atom Cu-N₄ supported on N doped carbon polyhedron**) > Pt/C.”

“**Preparation of S-Cu-ISA/SNC and the comparison samples.** In a typical synthesis of S-Cu-ISA/SNC catalyst, firstly the precursors were prepared by mixing sulfur powder and Cu-ZIF-8 (Supplementary Note. 1) in 20ml of mixture solution (carbon tetrachloride: ethanol = 4:1) under sonication. Then the mixture was heated to 60 °C with vigorous stirring and then stayed until drying. Afterwards, the dry powder was ground sufficiently and placed in the quartz tube, followed by heating at 450 °C in Ar for 2 hours, and then pyrolyzed in Ar at 950 °C for 4 hours. For the comparison samples, SNC (without the addition of Cu(acac)₂) was prepared as the same process. Cu-ISA/NC (**isolated single-atom Cu-N₄ supported on N doped carbon polyhedron**,

without the addition of S) and NC (N doped carbon polyhedron, without the addition of S and Cu) were obtained by pyrolysis of Cu-ZIF-8 and pure ZIF-8, respectively. The preparation of Cu-ISA/SNC (isolated single-atom Cu-N₄ supported on N and S co-doped carbon polyhedron) was described in Supplementary Note. 2.”

Our revision: We have modified the manuscript according to the reviewer’s suggestions (Line 99-102 and Line 500-504 in the revised manuscript).

Pg 6 line 134 : «N₂ absorption-desorption” => adsorption-desorption

Response: Thank you for your kind suggestions. We have carefully checked the language and modified mistakes in the revised manuscript (Line 138).

Pg 7, line 144 : the atomistic dispersion => atomic dispersion

Response: We have modified mistake in the revised manuscript (Line 148).

Figure caption of figure S9 is unclear: “size distribution of single-atom Cu” ...the title says that these are single atoms Cu, while the data is supposed to demonstrate that. Maybe the title could be “The size distribution of Cu signal detected by EDS-HRTEM”?

Response: Thank you for your kind suggestions. We have modified mistake in the revised Supplementary Material (Page 10).

Response to Reviewer 3:

The manuscript, entitled "Engineering unsymmetrical isolated Cu-S₁N₃ atomic interface for boosting oxygen reduction reaction ", reported that a N, S co-coordinated Cu based single atom electrocatalyst shows an excellent ORR performance. The authors claimed that this kind of asymmetric coordinate structure offers a more suitable free energy and favorable adsorption abilities for intermediates. The work is of significance and novelty in exploring the underlying catalytic mechanism of single metal atomic site such as Cu-S₁N₃ coordination structure. The experiments are reasonable, and the results are convincing. So I recommend this work to be published in Nature Communications. Before it can be accepted by the journal, there are a few questions needed to be clarified.

Response: Thank you for your positive comments on our manuscript. We have revised our manuscript according to your suggestions.

1. In Fig 1e, the scale bar should be provided.

Response: Thank you for your kind suggestions. We have provided the scale bar in the **Fig. R28**. Moreover, we also added scale bar in other STEM images in the manuscript and supplementary materials.

Fig. R28. Morphology and composition characterizations of S-Cu-ISA/SNC. (a) SEM, (b) TEM and (c) EDS images of S-Cu-ISA/SNC, C (pink), N (green), S (yellow) and Cu (red). (d) HAADF-STEM image and (e) the magnified image of S-Cu-ISA/SNC. (f) The corresponding intensity profiles along the line X-Y in e.

Our revision: We have modified the manuscript according to the reviewer's suggestions (**Fig. 1** in the revised manuscript, **Supplementary Fig. 14d**, **Supplementary Fig. 15d**, **Supplementary Fig. 42d**).

2. The ICP data of Zn species should be given to demonstrate its absence in the whole sample. If it exists, a further demonstration of its function in ORR performance should be clarified, as some recent work reported Zn-N₂ active sites may facilitate O-O bond activation (Nature Communications, 2019, DOI: 10.1038/s41467-019-10622-1).

Response: Thanks for the reviewer's comment. At one standard atmosphere, the boiling point of Zn metal is about 907 °C. Because the pyrolysis temperature of 950 °C is distinct higher than the boiling point of Zn metal, Zn metal will vaporize away along with the Ar flow at such high temperature and longtime of 4 hours.¹⁻³ The residual Zn content in the S-Cu-ISA/SNC is 0.028 at%, according to the inductively coupled plasma optical emission spectrometry (ICP-OES) analysis, which is much lower than that of Cu (0.73 at%) in the sample. In addition, we have carried out energy-dispersive spectroscopy (EDS) mappings and XPS measurements to confirm the existence of the elements in our S-Cu-ISA/SNC catalyst. As shown in **Fig. 1c** and **Supplementary Fig. 9**, Cu, S, N and C elements are distributed uniformly over the entire architecture of S-Cu-ISA/SNC catalyst, and almost no Zn related signals was observed, which may due to the ultralow content of Zn in S-Cu-ISA/SNC catalyst. From the XPS spectra for the survey scan of S-Cu-ISA/SNC in **Supplementary Fig. 16a**, no obvious Zn related peak was observed, further confirming the ultralow content of Zn in the sample. The atomic content percentages of C, N, S, Cu and Zn in S-Cu-ISA/SNC investigated by XPS or ICP measurement were shown in **Table R2**.

To investigate the influence of Zn species with ultralow content to the ORR performance, we have carefully examined the LSV analysis for the Zn-contained samples in 0.1M KOH. In **Fig. 3a** and **Fig. 3b (Fig. R29)**, particularly for NC and SNC (in **Table R3**, the Zn content in the two samples are 0.030 and 0.025 according to ICP measurements, which is almost the same to that of S-Cu-ISA/SNC), the $E_{1/2}$ value was 0.66 V and 0.79 V respectively, significantly lower than that of S-Cu-ISA/SNC (0.918 V), Cu-ISA/SNC (0.87 V) and Cu-ISA/NC (0.86 V). Furthermore, the catalytic activity of NC catalyst in this work is comparable to that of Zn

absence NC materials in previous works.^{4,5} The above results confirm that the residual Zn has little influence for ORR performance in the S-Cu-ISA/SNC system.

Table R2: The atomic content percentages of C, N, S, Cu and Zn in S-Cu-ISA/SNC measured by XPS or ICP analysis.

Element	C	N	S	Cu	Zn(ICP)
Content (at%)	87.82	9.5	2.05	0.61	0.028

Table R3: The atomic content percentages of the residual Zn in NC, SNC, Cu-ISA/NC and Cu-ISA/SNC measured by ICP analysis.

Element	NC	SNC	Cu-ISA/NC	Cu-ISA/SNC
Content (at%)	0.030	0.025	0.032	0.021

Fig. R29. (a) Polarization curves for S-Cu-ISA/SNC, Cu-ISA/SNC, Cu-ISA/NC, SNC, NC and 20% Pt/C. (b) Comparison of J_k at 0.85 V and $E_{1/2}$ for S-Cu-ISA/SNC and the reference catalysts.

References:

1. Liu, B., Shioyama, H., Akita, T., & Xu, Q. Metal-organic framework as a template for porous carbon synthesis. *J. Am. Chem. Soc.* **130**, 5390 (2008).
2. Chen, Y. Z., Wang, C., Wu, Z. Y., Xiong, Y., Xu, Q., Yu, S. H., & Jiang, H. L. From bimetallic metal-organic framework to porous carbon: high surface area and multicomponent active dopants for excellent electrocatalysis. *Adv. Mater.* **27**, 5010 (2015).
3. Yin, P., Yao, T., Wu, Y., Zheng, L., Lin, Y., Liu, W., Zhou, G. Single cobalt atoms with precise N-coordination as superior oxygen reduction reaction catalysts. *Angew. Chem. Int. Ed.* **55**, 10800 (2016).
4. Yang, Q., Jia, Y., Wei, F., Zhuang, L., Yang, D., Liu, J., Yao, X. Understanding the Activity of Co-N_{4-x}C_x in Atomic Metal Catalysts for Oxygen Reduction Catalysis. *Angew. Chem. Int. Ed.* **59**, 2-8 (2020).
5. Li, F., Bu, Y., Han, G. F., Noh, H. J., Kim, S. J., Ahmad, I., Zhong, Q., et al. Identifying the structure of Zn-N₂ active sites and structural activation. *Nat Commun.* **10**, 2623 (2019).

Our revision: We have revised our manuscript according to the reviewer's suggestion. (**Line 159-161** in the revised manuscript)

3. The XPS spectrums of S and N species should be provided to analyze the type of these two elements in sample, which can help the rationality of DFT model.

Response: Thanks for the reviewer's critical and insightful comment. As mentioned in our manuscript and supplementary materials, the Cu atoms are coordinated by S and N to form Cu-S and Cu-N bond mainly from the XAFS and XPS results. We agree with the reviewer's statement that the type of S and N species provide important information for assisting to constructing the DFT model (**Fig. R30**). Actually, when constructing the models for DFT calculations, we have

considered the type of S and N species, as shown in **Supplementary Fig. 16**. The N 1s spectra (**Supplementary Fig. 16e**) was classified to four types of N species, namely, 398.4 (pyridinic N), 399.1 (Cu-N), 400.2 (pyrrolic N) and 401.1 (graphitic N). The S 2p XPS spectra (**Supplementary Fig. 16f**) show three types of S species. The peak at 168.3 eV assigned to the sulfate species (C-SO_x), while the two peaks at 164.9 eV and 163.6 eV were associated to C=S-C and C-S-C bond, respectively. Importantly, the characteristic peaks at 163.9 eV, corresponding to the Cu-S bond, were observed, which could stem from the partial replacement of N atoms with S to form the Cu-S coordinations. All these results indicated that the atomically dispersed Cu possessed typical Cu-N and Cu-S dual coordinating environments. So we propose the unsymmetrical isolated Cu-S₁N₃ atomic interface structure for S-Cu-ISA/SNC.

Fig. R30. The proposed atomic structures schematics based on XPS and XAFS analysis.

Our revision: We analyze the XPS spectrums of S and N species in Supplementary Materials and have modified the manuscript according to the reviewer's suggestions (**Supplementary Fig. 16**).

4. Different kinds of S-M-S₁N₃ ISA/SNC (Mn, Fe, Co, Ni) samples have been synthesized successfully. How about their ORR performance as compared to Cu based N, S co-coordinated structure?

Response: We thank the reviewer for this comment. We evaluated the ORR performance of S-M-ISA/SNC (Mn, Fe, Co, Ni) samples in alkaline media in 0.1 M KOH solution. **Fig. R31** exhibited the LSV curves for S-Mn-ISA/SNC, S-Fe-ISA/SNC, S-Co-ISA/SNC and S-Ni-ISA/SNC. As we can see, the samples of S-M-ISA/SNC (M=Mn, Fe, Co, Ni) show optimistic performance. In **Fig. R32**, the half-wave potential ($E_{1/2}$) of S-Mn-ISA/SNC, S-Fe-ISA/SNC, S-Co-ISA/SNC and S-Ni-ISA/SNC was 0.902 V, 0.917 V, 0.911 V and 0.851 V, respectively. The favorable ORR kinetics of S-Mn-ISA/SNC, S-Fe-ISA/SNC, S-Co-ISA/SNC and S-Ni-ISA/SNC was verified by kinetic current density of 14.5 mA cm⁻², 40.0 mA cm⁻², 27 mA cm⁻² and 5.1 mA cm⁻². The Tafel slope of S-Mn-ISA/SNC, S-Fe-ISA/SNC, S-Co-ISA/SNC and S-Ni-ISA/SNC was calculated to be 83.8 mV dec⁻¹, 62.6 mV dec⁻¹, 72.5 mV dec⁻¹ and 91.7 mV dec⁻¹ (**Fig. R33**), further demonstrating the desirable ORR kinetics for S-M-ISA/SNC (M=Mn, Fe, Co, Ni).

Fig. R31. ORR polarization curves for S-Mn-ISA/SNC, S-Fe-ISA/SNC, S-Co-ISA/SNC and S-Ni-ISA/SNC.

Fig. R32. Comparison of J_k at 0.85 V and $E_{1/2}$, for S-Mn-ISA/SNC, S-Fe-ISA/SNC, S-Co-ISA/SNC and S-Ni-ISA/SNC.

Fig. R33. Tafel plots of S-Mn-ISA/SNC, S-Fe-ISA/SNC, S-Co-ISA/SNC and S-Ni-ISA/SNC in alkaline media.

Our revision: We have added the ORR performance of S-M-ISA/SNC (Mn, Fe, Co, Ni) samples in the revised manuscript. (Line 454-465 in the revised manuscript and **Supplementary Fig. 72-74** in the supplementary materials)

5. The asymmetry of the electronic redistribution discussed in *Adv Mater* 2019, 1805581 may offer help for discussion in this work.

Response: Thank you for the valuable suggestion. We read this paper carefully and strongly agree the viewpoint that the formation of hetero-coordinated moieties at atomic scale with asymmetrical electronic distribution should be conducive to the oxygen-involving catalysis, as it can benefit the adsorption and desorption of the adsorbates.

Our revision: We added the paper (*Adv. Mater.* **2019**, 1805581) as reference in the revised version (**Reference 44**).

Reviewers' Comments:

Reviewer #1:

Remarks to the Author:

I think that the most critical issue of this work was to demonstrate the existence of a Cu-S bond integrated in a CuN_xC_y moiety, and not attributable to other copper sulfide species. The authors have addressed the point in a satisfactory manner, and the inclusion of S K-edge XAS experiments improved and completed the study. I then recommend the publication of the manuscript, but I also suggest to remove the linear combination fitting part, because these fits have been performed with a calculated spectrum of CuN_4 . A computed spectrum is not a real data, and can introduce some artefacts.

Reviewer #2:

Remarks to the Author:

The authors have made tremendous efforts to answer all the questions of this reviewer, and added additional experimental results in the revised manuscript.

In particular, the linear combination fitting figures R20-R22 highlight that it is not possible to fit the xanes spectrum properly only with CuN_4 and Cu-sulphides, further giving indirect evidence of the presence of Cu single atoms coordinated with both N and S.

The manuscript can be accepted without further modification.

Reviewer #3:

Remarks to the Author:

The authors addressed all my concerns with satisfactory, so I recommend this manuscript publication in Nat Comm.

Responses to the Referees' Comments

We thank the referees for their valuable comments to our manuscript. We have carefully considered the referees' comments and revised the manuscript accordingly. Our responses and corresponding revisions are as follows:

Response to Reviewer 1:

I think that the most critical issue of this work was to demonstrate the existence of a Cu-S bond integrated in a CuN_xC_y moiety, and not attributable to other copper sulfide species. The authors have addressed the point in a satisfactory manner, and the inclusion of S K-edge XAS experiments improved and completed the study. I then recommend the publication of the manuscript, but I also suggest to remove the linear combination fitting part, because these fits have been performed with a calculated spectrum of CuN_4 . A computed spectrum is not a real data, and can introduce some artefacts.

Response: Thank you for your positive comments on our manuscript. As we know, ATHENA has a capability of fitting a linear combination of standard spectra to an unknown spectra. These fits can be done using normalized $\mu(\text{E})$ spectra. In fact, it is hard for us to obtain a standard experimental spectrum which is perfectly matched with the CuN_4 model, so a calculated spectrum of CuN_4 was applied to give indirect evidence of the presence of Cu single atoms coordinated with both N and S. For this reason, we retained the LCF part in the manuscript.

Our reversion: We revised the Supplementary Information (Page 29).

Response to Reviewer 2:

The authors have made tremendous efforts to answer all the questions of this reviewer, and added additional experimental results in the revised manuscript. In particular, the linear combination fitting figures R20-R22 highlight that it is not possible to fit the xanes spectrum properly only with CuN₄ and Cu-sulphides, further giving indirect evidence of the presence of Cu single atoms coordinated with both N and S. The manuscript can be accepted without further modification.

Response: We are very grateful to the reviewer for his/her support and recognition in this work.

Response to Reviewer 3:

The authors addressed all my concerns with satisfactory, so I recommend this manuscript publication in Nat Comm.

Response: Many thanks to the reviewer for appreciating to this work.